# 2-oxoglutarate triggers assembly of active dodecameric *Methanosarcina mazei* glutamine synthetase

Eva Herdering[1†], Tristan Reif-Trauttmansdorff[2†], Anuj Kumar[2], Tim Habenicht[1], Georg Hochberg[2,3,4], Stefan Bohn[5], Jan Schuller[2]*, Ruth Anne Schmitz[1]*

[1]Institute for General Microbiology, Christian Albrechts University, Kiel, Germany; [2]Center for Synthetic Microbiology (SYNMIKRO) Research Center and Department of Chemistry, Philipps-Universität Marburg, Marburg, Germany; [3]Evolutionary Biochemistry Group, Max Planck Institute for Terrestrial Microbiology, Marburg, Germany; [4]Department of Chemistry, Philipps-Universität Marburg, Marburg, Germany; [5]Cryo-Electron Microscopy Platform and Institute of Structural Biology, Helmholtz Munich, Neuherberg, Germany

*For correspondence:
jan.schuller@synmikro.uni-marburg.de (JS);
rschmitz@ifam.uni-kiel.de (RAS)

[†]These authors contributed equally to this work

Competing interest: The authors declare that no competing interests exist.

## eLife Assessment

This study reveals a novel mechanism of glutamine synthetase (GS) regulation in Methanosarcina mazei, demonstrating that 2-oxoglutarate (2-OG) directly promotes GS activity by stabilizing its dodecameric assembly. Using mass photometry, activity assays, and cryo-electron microscopy, the authors show that GS transitions from a dimeric, inactive form at low 2-OG concentrations to a fully active dodecameric complex at saturating 2-OG levels, highlighting 2-OG as a key effector in C/N sensing. The findings are **valuable**, supported by **solid** data, and provide new insights into archaeal GS regulation, though further clarification of interactions with known partners like Glnk1 and sp26 is needed.

**Abstract** Glutamine synthetases (GS) are central enzymes essential for the nitrogen metabolism across all domains of life. Consequently, they have been extensively studied for more than half a century. Based on the ATP-dependent ammonium assimilation generating glutamine, GS expression and activity are strictly regulated in all organisms. In the methanogenic archaeon *Methanosarcina mazei*, it has been shown that the metabolite 2-oxoglutarate (2-OG) directly induces the GS activity. Besides, modulation of the activity by interaction with small proteins ($GlnK_1$ and sP26) has been reported. Here, we show that the strong activation of *M. mazei* GS ($GlnA_1$) by 2-OG is based on the 2-OG dependent dodecamer assembly of $GlnA_1$ by using mass photometry (MP) and single particle cryo-electron microscopy (cryo-EM) analysis of purified strep-tagged $GlnA_1$. The dodecamer assembly from dimers occurred without any detectable intermediate oligomeric state and was not affected in the presence of $GlnK_1$. The 2.39 Å cryo-EM structure of the dodecameric complex in the presence of 12.5 mM 2-OG demonstrated that 2-OG is binding between two monomers. Thereby, 2-OG appears to induce the dodecameric assembly in a cooperative way. Furthermore, the active site is primed by an allosteric interaction cascade caused by 2-OG-binding towards an adaption of an open active state conformation. In the presence of additional glutamine, strong feedback inhibition of GS activity was observed. Since glutamine dependent disassembly of the dodecamer was excluded by MP, feedback inhibition most likely relies on the binding of glutamine to the catalytic site. Based on our findings, we propose that under nitrogen limitation the induction of *M. mazei* GS into a catalytically active dodecamer is not affected by $GlnK_1$ and crucially depends on the presence of 2-OG.

## Introduction

Nitrogen is one of the key elements in life and it is essentially required in the form of ammonium for biomolecules such as proteins or nucleic acids. Two major pathways of ammonium assimilation in bacteria and archaea are known. Under nitrogen (N) sufficiency, glutamate dehydrogenase (GDH) is active and generates glutamate from 2-OG and ammonium (reviewed in *van Heeswijk et al., 2013*). Under N limitation however, low ammonium concentrations lead to an inactive GDH as a result of its low ammonium affinity, whereas the expression of GS is strongly induced in response to N limitation (*Bolay et al., 2018*; *Gunka and Commichau, 2012*; *Stadtman, 2001*). Consequently, under low ammonium conditions, GS together with glutamate synthase (GOGAT) are responsible for ammonium assimilation via the GS/GOGAT pathway, one of the major intersections in central carbon and N metabolism. Accordingly, GS present across all domains of life plays a central role in cellular N assimilation under low N availability. The enzyme, its structure, and regulation have been investigated in detail in different organisms for more than half a century (e.g. *dos Santos Moreira et al., 2019*; *Stadtman, 2001*; *Woolfolk and Stadtman, 1967*).

Most of the GS are grouped into three major classes based on their monomeric size and oligomerization properties (overview in *dos Santos Moreira et al., 2019*). GSI and GSIII, both found in bacteria and archaea, mostly form dodecamers, whereas GSII found in Eukaryotes form decamers of smaller subunits (*dos Santos Moreira et al., 2019*; *He et al., 2009*; *Valentine et al., 1968*; *van Rooyen et al., 2011*). The GSI class can be further grouped into Iα-type GS and Iβ-type GS based on their amino acid sequence and respective molecular mechanisms of activity regulation. Iβ-type GS contain a conserved adenylylation site (Tyr397 residue near the active site), that allows for covalent modification of Iβ-type GS and leads to inactivation of the enzyme (*Brown et al., 1994*; *Magasanik, 1993*; *Shapiro and Stadtman, 1970*). Iα-type GS on the other hand are not covalently modified and mainly show feedback inhibition by end products of the glutamine metabolism, including glutamine (*Fisher, 1999*; *Gunka and Commichau, 2012*).

## GS regulation on transcriptional level

Since in contrast to GDH, GS-catalyzed generation of glutamine requires ATP, most organisms strictly regulate the expression of GS in response to the nitrogen availability on the transcriptional level. In gram-negative bacteria, mainly transcriptional activation of the coding gene (*glnA*) under low nitrogen availability occurs via a transcriptional activator (e.g. NtrC in *Escherichia coli Jiang et al., 1998*). For several gram-positive bacteria however, the mechanism of regulation is a de-repression of *glnA* transcription under N limitation, which has also been shown for methanoarchaea (*Cohen-Kupiec et al., 1999*; *Fedorova et al., 2013*; *Fisher, 1999*; *Fisher and Wray, 2008*; *Hauf et al., 2016*; *Weidenbach et al., 2010*; *Weidenbach et al., 2008*). Whereas in gram positives the signal perception is complex and often also involves protein interactions of GS with transcriptional regulators (reviewed in *Gunka and Commichau, 2012*), signal perception and transduction in methanoarchaea occurs directly via the small effector molecule 2-OG, which increases under N limitation. It has been shown that binding of 2-OG to the global N repressor protein NrpR significantly changes the repressor conformation resulting in dissociation from its respective operator (*Lie et al., 2007*; *Weidenbach et al., 2010*; *Wisedchaisri et al., 2010*). In addition to expression regulation, the activity of GS is also strictly regulated in all organisms in response to changing N availabilities, however, the underlying molecular mechanism(s) of inhibition significantly differ for the various GS classes and in various organisms (*Reitzer, 2003*).

## Regulation of GS activity: Highly diverse and often complex in various organisms

An extensive repertoire of cellular control mechanisms regulating GS activity in response to N availability has been observed in different organisms. Inhibitory mechanisms in response to an N upshift range from feedback inhibition by e.g. glutamine or other end products of the glutamine metabolism (e.g. *E. coli* (*Stadtman, 2004*), *Bacillus subtilis* (*Deuel et al., 1970*), yeast *Legrain et al., 1982*), proteolytic degradation (yeast, *Legrain et al., 1982*), covalent modification by adenylylation of the Iβ-type GS subunits (e.g. enterobacteriaceae), thiol-based GS regulation (e.g. in soybean nodules *Masalkar and Roberts, 2015*), inhibition by regulatory proteins (e.g. in gram-positive bacteria *Travis et al., 2022*), inhibition by interactions with small proteins (e.g. inhibitory

factors in cyanobacteria *García-Domínguez et al., 1999*; *Klähn et al., 2015*, *Klähn et al., 2018*), to directly effecting the activity through the presence or absence of the small metabolite 2-OG, which has been shown for the first time for *Methanosarcina mazei* (*Ehlers et al., 2005*). Moreover, often several of the different regulatory mechanisms for GS activity are reported for one organism. For example, yeast GS (ScGS) is regulated via feedback inhibition by glutamine and additionally is susceptible to proteolytic degradation under N starvation. It was also found to assemble into nano-tubes (*He et al., 2009*) and under advanced cellular starvation into inactive filaments (*Petrovska et al., 2014*). In *E. coli*, the activity of the Iß-type GS (EcGS) is controlled by cumulative feedback inhibition and covalent modification (reviewed in *Reitzer, 2003*). It has been shown that each of the 12 subunits can be modified by adenylylation (Tyr397) resulting in an inactivation of the respective subunit (*Stadtman, 1990*). Moreover, the adenylylation of single subunits makes the other subunits more susceptible to cumulative feedback inhibition by various substances (*Stadtman, 1990*). These substances either bind the glutamine-binding pocket or have an allosteric binding site (*Liaw et al., 1993*; *Woolfolk and Stadtman, 1967*). The dodecameric structure of EcGS has been known for a long time (*Almassy et al., 1986*; *Yamashita et al., 1989*). However, when artificially exposed to divalent cations ($Mn^{2+}$, $Co^{2+}$), it randomly aggregates and produces long hexagonal tubes (paracrystalline aggregates) (*Valentine et al., 1968*). The detailed structural information on the mechanisms of this reversible GS-filament formation to an inactive form of EcGS, often associated with stress responses, has only recently been described by cryo-electron microscopy (cryo-EM) analysis (*Huang et al., 2022*). The *B. subtilis* GS has been shown to be feedback regulated. In addition, binding of the transcriptional repressor GlnR to the feedback inhibited complex not only activates the transcription repression function of GlnR (*Fisher and Wray, 2008*), but also stabilizes the inactive GS conformation potentially changing from a dodecamer into a tetradecameric structure (*Travis et al., 2022*).

In *M. mazei*, a mesophilic methanoarchaeon which is able to fix $N_2$, regulation of the central N metabolism has been studied extensively on the transcriptional and post-transcriptional level (*Jäger et al., 2009*; *Prasse and Schmitz, 2018*; *Veit et al., 2005*). A central role of 2-OG for the perception of changes in N availabilities has been proposed, as has been demonstrated for cyanobacteria (*Forchhammer, 1999*; *Herrero et al., 2001*). The activity of *M. mazei* GS, encoded by $glnA_1$, is regulated by several different mechanisms. $GlnA_1$ is not covalently modified in response to N availability and thus represents an Iα-type-GS (*Ehlers et al., 2005*). It has been proposed, that $GlnA_1$ is directly activated under N starvation by the high intracellular concentrations of the metabolite 2-OG (*Ehlers et al., 2005*). 2-OG represents the internal signal for N limitation, since the internal 2-OG level significantly increases due to missing consumption by GDH under N starvation (*M. mazei* contains the oxidative TCA part, anabolic). The increased cellular 2-OG concentration has been shown to be directly perceived by $GlnA_1$, most likely by direct binding resulting in strong activation (*Ehlers et al., 2005*). Besides, we showed first evidence that two small proteins interact with *M. mazei* $GlnA_1$, the PII-like protein $GlnK_1$ and small protein sP26 comprising 23 amino acids (*Ehlers et al., 2005*; *Gutt et al., 2021*). The presence and potential interaction of both small proteins showed small effects on the $GlnA_1$ activity. However, those small effects might be neglectable compared to the strong 2-OG stimulation, particularly taking into account that the indirect GS activity assay shows high deviations in the low activity range. Moreover, initial complex formation analysis by a pull-down approach indicated that in the absence of 2-OG, the $GlnA_1/GlnK_1$ complexes are more stable than in the presence of high 2-OG. This led to the conclusion that due to the shift to N sufficiency after a period of N limitation, $GlnA_1$ activity is reduced due to the lower 2-OG concentration, but also due to a potential inhibitory protein interaction with $GlnK_1$ (*Ehlers et al., 2005*). Very recently, the first structural analysis of *M. mazei* $GlnA_1$ was reported, showing GS complexes with $GlnK_1$ (*Schumacher et al., 2023*). Based on their findings, Schumacher et al. propose a regulation of $GlnA_1$ activity by oligomeric modulation, with $GlnK_1$ stabilizing the dodecameric structure and the formation of $GlnA_1$ active sites. Since that work is entirely missing the effects of 2-OG on $GlnA_1$ structure and activity, we here aimed to study the regulation of *M. mazei* $GlnA_1$ in more detail by evaluating oligomerization and complex formation between $GlnA_1$, $GlnK_1$ and sP26 in dependence of 2-OG. This was achieved by employing mass photometry (MP), allowing the measurement of the molecular weight distribution of single complexes in solution, and by high resolution cryo-EM, whilst also performing activity assays.

## Results

### 2-OG is responsible for GlnA$_1$-dodecamer formation in *M. mazei*

The strep-tagged purified GlnA$_1$ was analyzed by size exclusion chromatography (SEC) in the presence of 12.5 mM 2-OG, demonstrating that GS is exclusively present in a dodecameric structure, no other oligomers were detectable (*Figure 1—figure supplement 2*). To investigate the effects of 2-OG on *M. mazei* GlnA$_1$ in more detail, we employed MP, a method that allows to measure the molecular weight distribution of particles in solution. Strep-tagged purified GlnA$_1$ (after SEC) was dialyzed into a 2-OG free HEPES buffer (see Materials and Methods) and subsequently analyzed by MP, demonstrating that in the presence of low 2-OG concentrations (0.1 mM) all of the *M. mazei* GlnA$_1$ was nearly exclusively present as dimers with no higher molecular weight complexes present. After the addition of 12.5 mM 2-OG, the size distribution shifted towards a higher molecular weight complex of 630–700 kDa (calculated based on the measured dimer size in each measurement; expected molecular weight of dodecamer: 634 kDa) (*Figure 1A and B*). This molecular weight corresponds to a fully assembled dodecamer species, the same oligomeric structure that is adapted in GS from other prokaryotes. Using 2-OG concentrations varying between 0.1 and 12.5 mM, complex analysis showed that up to 62% of all particles were assembled in a dodecamer. This allowed to determine the effective concentration of 2-OG for dodecamer assembly to be EC50=0.75 ± 0.01 mM 2-OG (based on two biological replicates, calculated with the percentage of dodecamer) as described in Materials and Methods, and further verified that no other intermediate oligomeric complexes were detectable during dodecameric assembly (*Figure 1A*, *Figure 1—figure supplement 1A, B*). Notably, GlnA$_1$ did not reach 100% dodecamer assembly after removal and re-addition of 2-OG, although only dodecameric GlnA$_1$ was used for dialysis (*Figure 1—figure supplement 2B, C*). We conclude that GlnA$_1$ is rather unstable in the absence of 2-OG and some of the protein lost its ability to oligomerize after 2-OG was removed by dialysis.

Furthermore, 2-OG did not only cause dodecamer assembly but also higher enzyme activity. Activity measurements of Strep-GlnA$_1$ in the presence of increasing 2-OG concentrations showed a strong increase from 0.0 U/mg in the absence of 2-OG up to 7.8 ± 1.7 U/mg in the presence of 12.5 mM 2-OG (six independent protein purifications). The EC50 for GlnA$_1$ activity was determined to be 6.3 mM 2-OG (*Figure 1D*). Thus, we conclude that 2-OG first acts as a trigger for dodecameric assembly of *M. mazei* GlnA$_1$ (with an EC50=0.75 mM 2-OG), setting it apart from other bacterial and eukaryotic enzyme variants. Moreover, most likely in addition to the dodecameric assembly, 2-OG is required for a further 2-OG-induced conformational switch of the active site, since saturated GlnA$_1$ activities are not reached in the presence of 5 mM 2-OG, when most of the GlnA$_1$ is in a dodecameric structure. For full activity, the presence of at least 12.5 mM 2-OG is required (EC50=6.3 mM 2-OG).

### GlnK$_1$ has no detectable effects on GS dodecamer assembly or activity under the tested conditions

Previous studies have shown protein interactions between *M. mazei* GlnA$_1$ and GlnK$_1$ as well as GlnK$_1$ induced effects on GlnA$_1$ activity (ratio GlnA$_1$:GlnK$_1$ 1:1, (*Ehlers et al., 2005*) and 2:1.4 (*Gutt et al., 2021*) in activity assays). Consequently, we next tested the effects of GlnK$_1$ on GlnA$_1$ oligomerization in the presence of 2-OG. Performing the MP analysis under the tested conditions as before but in the presence of purified GlnK$_1$, demonstrated that (i) in the absence of 2-OG varying ratios between GlnA$_1$ and GlnK$_1$ (20:1, 2:1, 2:10 calculated based on monomer mass) did not result in any dodecamer assembly of GlnA$_1$ (*Figure 2A and B*), (ii) no difference in the GlnA$_1$ dodecameric assembly in the presence of 2-OG was obtained in the presence of purified GlnK$_1$ (ratio 2:1), (iii) nor was binding of GlnK$_1$ to GlnA$_1$ detected by a respective increase in the mass of the higher oligomeric complex (*Figure 2B and C*). Moreover, the presence of GlnK$_1$ (ratio 2:1) neither had an influence on the 2-OG affinity (EC50 (- GlnK$_1$)=1.06 mM 2-OG; EC50 (+ GlnK$_1$)=1.02 mM 2-OG, EC50 calculated based on the dodecamer/dimer ratio), nor in any ratio on the specific activity of GlnA$_1$ (*Figure 2D and E*: exemplarily showing 2:1; *Figure 1—figure supplement 1C*, D). Consequently, we conclude that under the conditions tested using purified proteins, GlnA$_1$ dodecamer assembly occurs independently of GlnK$_1$ and no binding of GlnK$_1$ to the dodecameric GlnA$_1$ occurs. However, we cannot exclude that cellular components/metabolites not present in these experiments are crucial for a GlnA$_1$-GlnK$_1$ interaction.

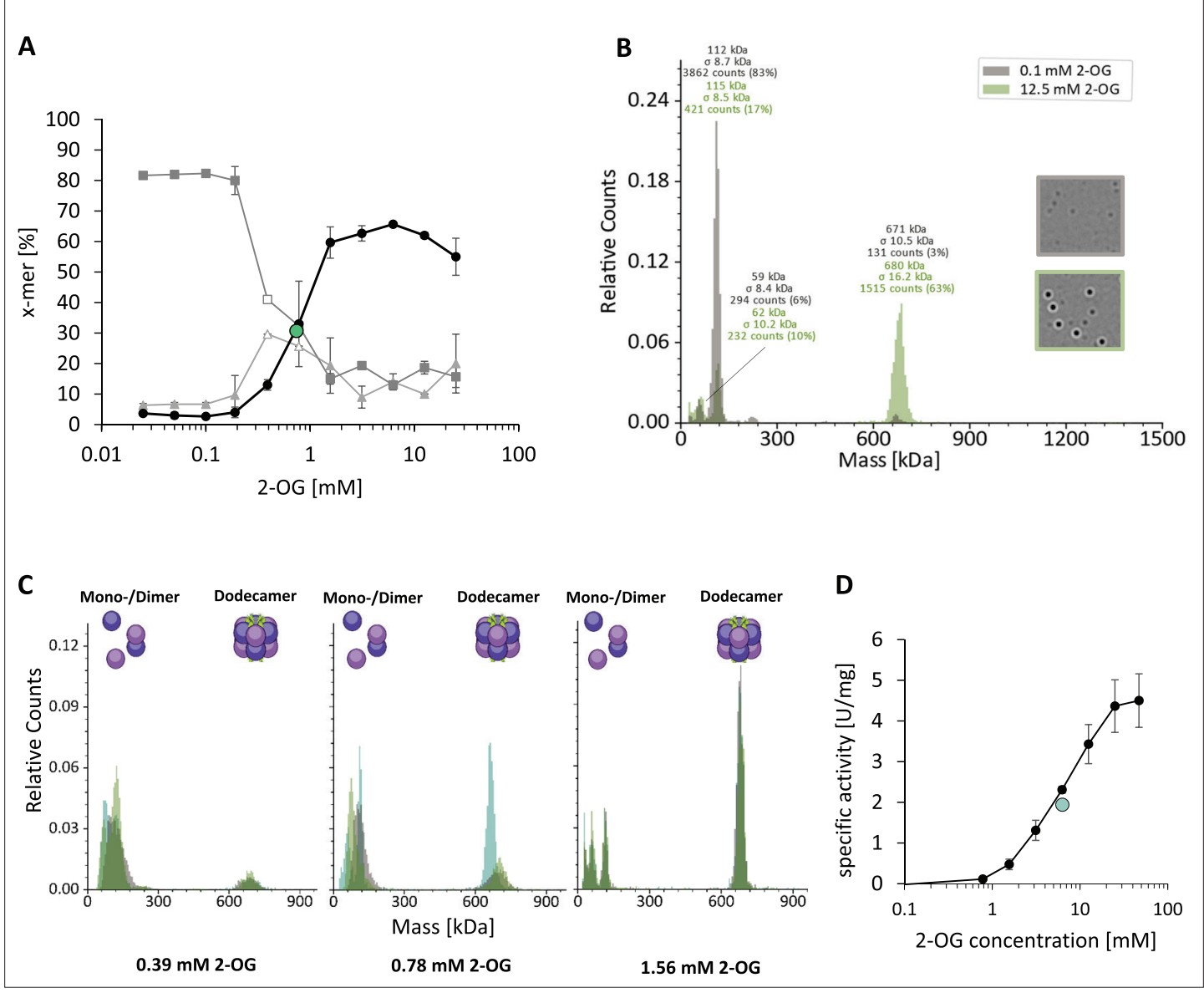

**Figure 1.** GlnA$_1$-dodecamer assembly is induced by 2-oxoglutarate (2-OG) without detectable oligomeric intermediates. Oligomerization states of purified strep-tagged GlnA$_1$ were assessed in dependence of 2-OG by mass photometry as described in MM using a Refeyn TwoMP mass photometer (Refeyn Ltd., Oxford, UK). Mass spectra are shown with relative counts (number of counts per peak in relation to the total number of counts) plotted against the molecular weight. (**A**) 75 nM GlnA$_1$ were preincubated in the presence of varying 2-OG concentrations (0–25 mM) for ten min at room temperature and kept on ice until measurement. The percentage of monomer (▲), dimer (■), and dodecamer (●) considering the total number of counts was plotted against the 2-OG concentration. One out of two independent biological replicates with each of three technical replicates is shown exemplarily and the EC50 for dodecamer assembly is indicated in green. Monomer and dimer-peaks were difficult to distinguish in the measurements for 0.39 and 0.78 mM 2-OG and the values are, therefore, shown without standard deviation and in white. (**B**) Exemplary mass spectra of GlnA$_1$ oligomers in the presence of 0.1 and 12.5 mM 2-OG. The molecular masses shown above the peaks correspond to a Gaussian fit of the respective peak (Gaussian fit not shown). (**C**) Mass spectra of the three technical replicates (different green colors) of GlnA$_1$-oligomers at 0.39, 0.78, and 1.56 mM 2-OG, excluding the presence of intermediates. (**D**) The specific activity of purified strep-tagged GlnA$_1$ was determined as described in MM in the presence of varying 2-OG concentrations (0, 0.78, 1.56, 3.13, 6.25, 12.5, 25, and 47 mM). The EC50 for GlnA$_1$-activity is shown in green and the standard deviation of four technical replicates is depicted.

The online version of this article includes the following source data and figure supplement(s) for figure 1:

**Figure supplement 1.** Sigmoidal fitted curves for mass photometry (**A-D**) and activity (**E**) measurements of Strep-GlnA$_1$ with varying concentrations of 2-oxoglutarate (2-OG).

*Figure 1 continued on next page*

Figure 1 continued

**Figure supplement 2.** Affinity-purified Strep-GlnA$_1$ and size-exclusion-chromatography (SEC) and mass photometry (MP) of Strep-GlnA$_1$ after purification.

**Figure supplement 2—source data 1.** 1.5 µg (lane 1) and 3 µg (lane 2) Strep-GlnA1 on a coomassie-stained 12 % SDS-Gel.

**Figure supplement 2—source data 2.** Size-exclusion-chromatography (SEC)-fractions of Strep-GlnA1 SEC-run; elution volume 1: 13.5-14 ml, 2: 14-15 ml, 3: 15-15.5 ml, 4: 19.5 – 20 ml on a coomassie-stained 12% SDS-Gel.

**Figure supplement 3.** Mass photometry of purified and thawed Strep-GlnA$_1$ before and after size-exclusion-chromatography (SEC).

**Figure supplement 4.** Original kinetic assay of Strep-GlnA$_1$ in the presence of 12.5 mM 2-oxoglutarate (2-OG).

## Structural basis of oligomer formation by 2-OG

To now unravel the structural mechanism underlying *M. mazei* GlnA$_1$ activation by 2-OG, we employed cryo-EM and single-particle analysis. Treating freshly purified Strep-GlnA$_1$ with 12.5 mM 2-OG, effectively shifted the equilibrium towards fully assembled homo-oligomers as depicted in the MP experiments (*Figure 1—figure supplement 2C*). In the micrographs, fully assembled ring-shaped particles are visible. However, initial attempts to obtain a 3D reconstruction were hindered by the pronounced preferred orientation of particles within the ice, a challenge which has been overcome by introducing low concentrations of CHAPSO (0.7 mM). In our final dataset, all particles exhibited well-distributed oligomers in diverse orientations. Leveraging this dataset, we aligned the particles to a 2.39 Å resolution structure, revealing well-resolved side chains that facilitated seamless model building (*Figure 3*, *Figure 3—figure supplement 1*, *Supplementary file 2*). Consequently, we achieved a structure demonstrating excellent geometry and density fitting.

The detailed structural analysis uncovered that GlnA$_1$ assembles into a dodecamer characterized by stacked hexamer rings. A single GlnA$_1$ protomer is composed of 15 β-strands and 15 α-helices and is split into a larger C-domain and an N-domain by helix α3. The dodecameric arrangement is achieved through two distinct interfaces, the hexamer interfaces and inter-hexamer interfaces. Hexamer interfaces are situated between subunits within each ring, while inter-hexamer interfaces occur between subunits derived from adjacent rings (*Figure 4A, B and C*). The structures are highly similar to Gram-positive bacterial GS structures (PDB: 4lnn, *Murray et al., 2013*), with root mean squared deviations (rmsds) of 0.5–1.0 Å.

A closer inspection of the density reveals the density for the bound 2-OG at an allosteric site localized at the interface between two GlnA$_1$ protomers in vicinity of the GlnA$_1$ catalytic site (*Figure 4B and D*). Several residues are contributing to its binding. R172' and S189' coordinate the γ-Carboxy-group. Additionally, two tightly bound water molecules are detectable in the binding site. One is interacting with the γ-Carboxy group, while being stabilized by another water that is coordinated by S38 and R26. Latter arginine is coordinating the α-Keto-group and, together with R87 and R173', the α-Carboxy group of 2-OG (*Figure 4B*). Notably, F24 stabilizes the 2-OG via stacking with its phenyl ring (*Figure 5*). This binding contribution from two GlnA$_1$ protomers at the intersubunit junction enhances activation by boosting readiness and the rate of full complex assembly. It operates akin to molecular glue that facilitate the observed cooperative assembly.

A comparison with the substrate-bound GlnA$_1$ structure (PDB: 8tfk, *Schumacher et al., 2023*) revealed that the catalytically important residues in *M. mazei* are the aspartic acid (D57), that abstracts the proton from ammonium, and the catalytic glutamic acid, Glu307. The active site of *M. mazei* GlnA$_1$ is formed at the interface between two subunits in the hexamer and formed by five key catalytic elements surrounding the active site: the E flap (residues 303–310), the Y loop (residues 369–377), the N loop (residues 235–247), the Y* loop (residues 152–161) and the D50' loop (residues 56–71). The latter one is the only one that originates from adjacent neighboring protomer (*Figure 5C and E*).

Superposition of our structure with the apo-*M. mazei* structure (PDB: 8tfb, *Schumacher et al., 2023*) reveals that 2-OG binding also triggers further movements that lead to structural changes in the substrate binding pocket (*Figure 5A, B and D*). R87' and its loop undergo a dramatic flip to coordinate 2-OG and D170 of helix α3 (residues 167–181) (*Figure 5A*). This, combined with the action of other coordinating residues, initiates a motion that is propagated through the entire protein. Notably, helix α3 shifts forward, causing F184 to flip over and facilitate a T-shaped aromatic interaction with F202. The resulting pull on F202 causes F204 to flip, allowing π-stacking with the purine moiety of ATP (*Figure 5B*). This series of structural changes primes the active site for ATP binding by already

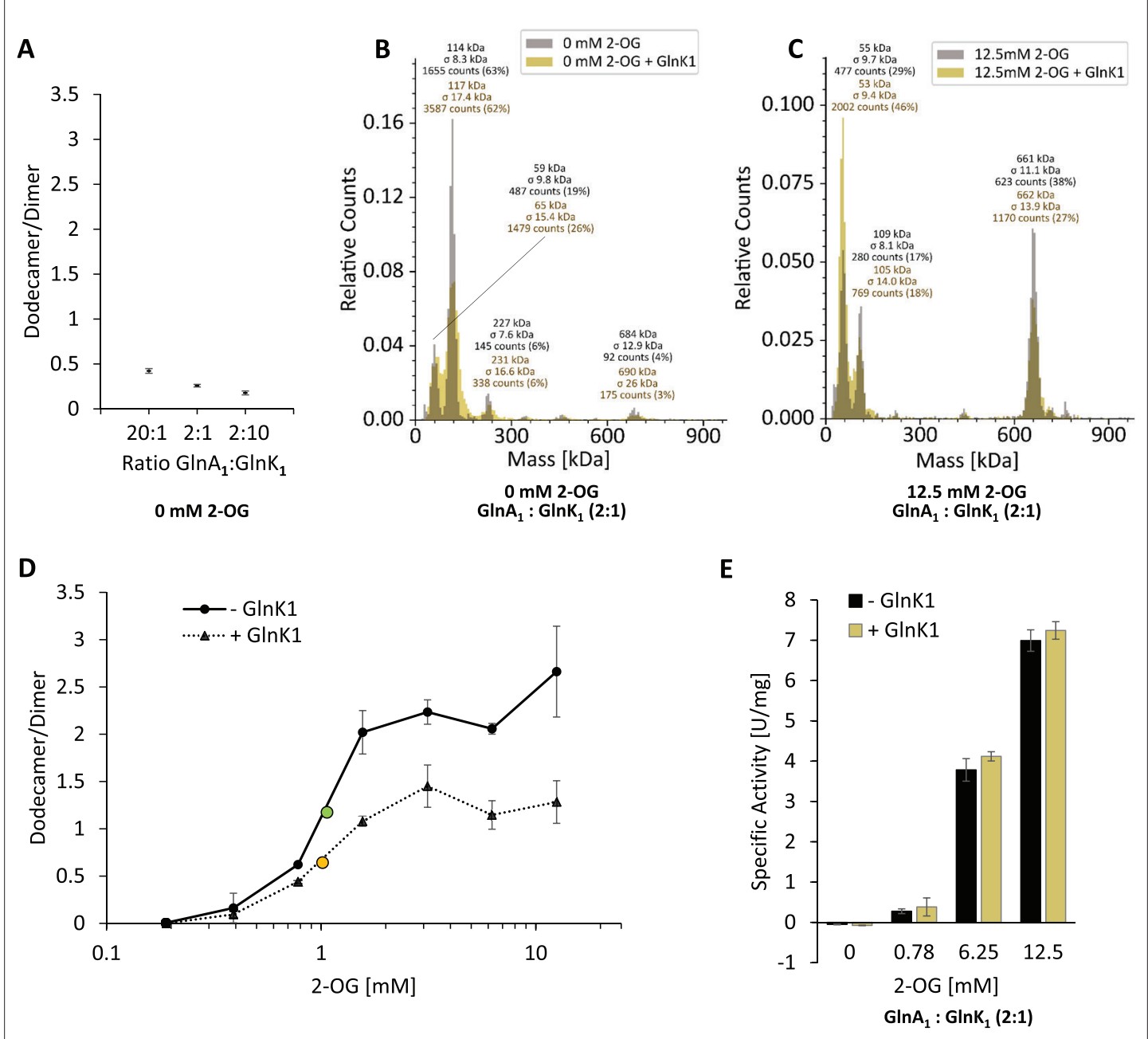

**Figure 2.** GlnA$_1$-dodecamer assembly and activity are not influenced by GlnK$_1$ under the conditions tested. Purified strep-tagged GlnA$_1$ and tag-less GlnK$_1$ were incubated in the absence or presence of 2-oxoglutarate (2-OG) in varying concentrations for 10 min at RT. Oligomerization states were assessed by mass photometry. Mass spectra are shown with relative counts (see **Figure 1**). (**A**) The obtained ratio of GlnA$_1$ dodecamer/dimer of three technical replicates are shown for varying ratios between GlnA$_1$ and GlnK$_1$ (20:1, 2:1, 2:10, ratios relating to monomers) in the absence of 2-OG. (**B, C**) Exemplary mass spectra of GlnA$_1$ incubated in the absence and presence of GlnK$_1$ (2:1) at 2-OG concentrations of 0 mM (B) and 12.5 mM (C). The molecular masses shown above the peaks correspond to a Gaussian fit of the respective peak (Gaussian fit not shown). (**D**) 200 nM GlnA$_1$ (molarity calculated based on molecular mass of monomers) were preincubated with GlnK$_1$ (in a 2:1 ratio) in the presence of varying 2-OG concentrations (0.19–12.5 mM) for ten min at RT. One biological replicate with three technical replicates was performed. The ratio of GlnA$_1$ dodecamer/dimer was plotted against the 2-OG concentration and the EC50 is indicated in green (●, - GlnK$_1$) and yellow (●, + GlnK$_1$). (**E**) The specific activity of purified strep-tagged GlnA$_1$ in the absence and presence of GlnK$_1$ (ratio 2:1) was determined as described in MM in the presence of varying 2-OG concentrations (0, 0.78, 6.25, and 12.5 mM). The standard deviations of four technical replicates of one biological replicate are indicated.

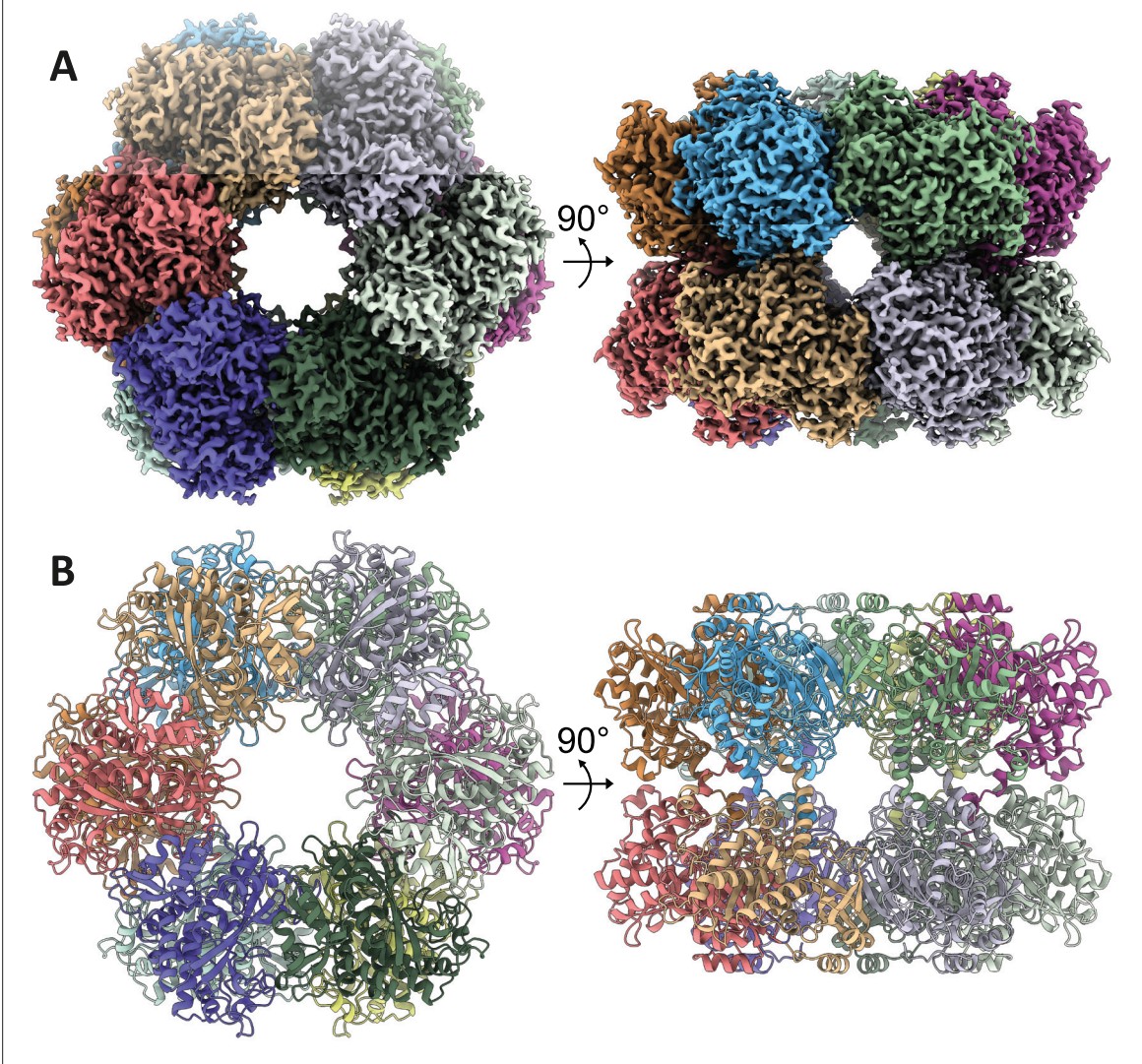

**Figure 3.** Structure of *M.Mazei* GlnA₁ with 2-oxoglutarate (2-OG). (**A**) Three-dimensional segmented cryo-electron microscopy (cryo-EM) density of the dodecameric complex colored by subunits. (**B**) Corresponding views of the GlnA₁ atomic model in cartoon representation.

The online version of this article includes the following figure supplement(s) for figure 3:

**Figure supplement 1.** Cryo-electron microscopy (Cryo-EM) Data processing workflow.

**Figure supplement 2.** *M. mazei* GlnA₁ filaments.

adopting the side chain conformations that are observed in analog (Met-Sox-P-ADP)-bound structure (transition state) (PDB: 8tfk, *Schumacher et al., 2023*), thus facilitating nucleotide binding (*Figure 5C and E*).

Additionally, the D50' loop adopts a position similar to the transition state in a catalytic competent conformation. This involved a remodeling of the loop, leading to the positioning of key catalytic residues in a catalytic competent configuration. Compared to the apo structure (*Schumacher et al., 2023*), R66 flips out of the catalytic pocket, now accommodating R319 which participates in phosphoryl transfer catalysis (*Liaw and Eisenberg, 1994*; *Figure 5D*). In addition, Asp 57' moves deeper into the binding site, facilitating the proton abstraction of $NH_4^+$ and preparing for its attack on the phosphorylated glutamate. Similar to the ATP/ADP binding site, these catalytic elements are primed to ideally stabilize the tetrahedral open active state. This is illustrated by the superposition of the inhibitor-bound, transition-state locked structure (*Schumacher et al., 2023*; *Figure 5C and E*).

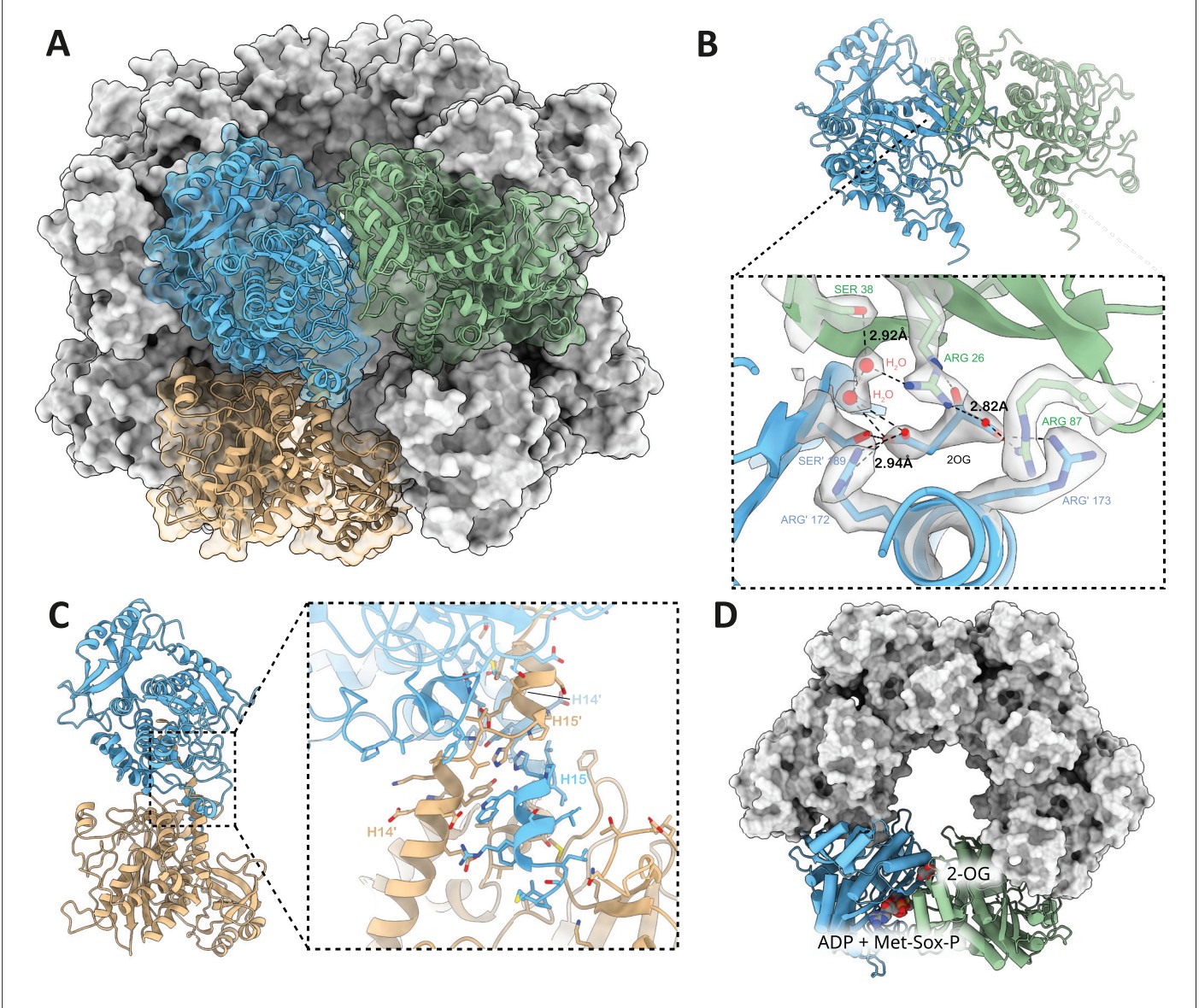

**Figure 4.** Hexameric interface, inter-hexameric interface, and 2-oxoglutarate (2-OG) binding site of dodecameric GlnA$_1$. (**A**) Surface representation of the *M. mazei* GlnA$_1$ 2-OG dodecamer with three GlnA$_1$ protomers fitted in cartoon representation into the dodecamer as dimers of inter-hexameric (blue and ochre) and hexameric (blue and green) GlnA$_1$. (**B**) Horizontal dimers and close-up of 2-OG binding site. Important residues are shown as atomic stick representation, primed labels indicate neighboring protomer. 2-OG and water molecules important for ligand binding fitted into density are shown in gray. Dotted lines represent polar interactions between 2-OG, waters, and residues. (**C**) Vertical dimers and close-up of dimerization site. C-terminal helices H14/15 and H14'/ H15' of two neighboring protomers lead to tight interaction, mediated by hydrophobic and polar interactions. (**D**) Top-view of GlnA$_1$ hexamer, 2-OG, and substrate binding sites are depicted for one horizontal dimer.

## Feedback inhibition by glutamine does not affect the dodecameric *M. mazei* GlnA$_1$ structure

For bacteria it is known, that GS can be feedback-inhibited. Very recently, the first feedback inhibition of an archaeal GS by glutamine has been reported for *Methermicoccus shengliensis* GS (*Müller et al., 2024*). The specific arginine residue identified to be relevant for the feedback inhibition is R66. Consequently, we generated the respective *M. mazei* GlnA$_1$ mutant protein changing the conserved arginine to alanine (R66A) (see also *Figure 5D and E*) and compared the purified strep-tagged mutant protein with the wildtype (wt) protein. In the presence of 12.5 mM 2-OG, the mutant protein showed the same specific activity as obtained for the wt. However, when supplementing 5 mM glutamine,

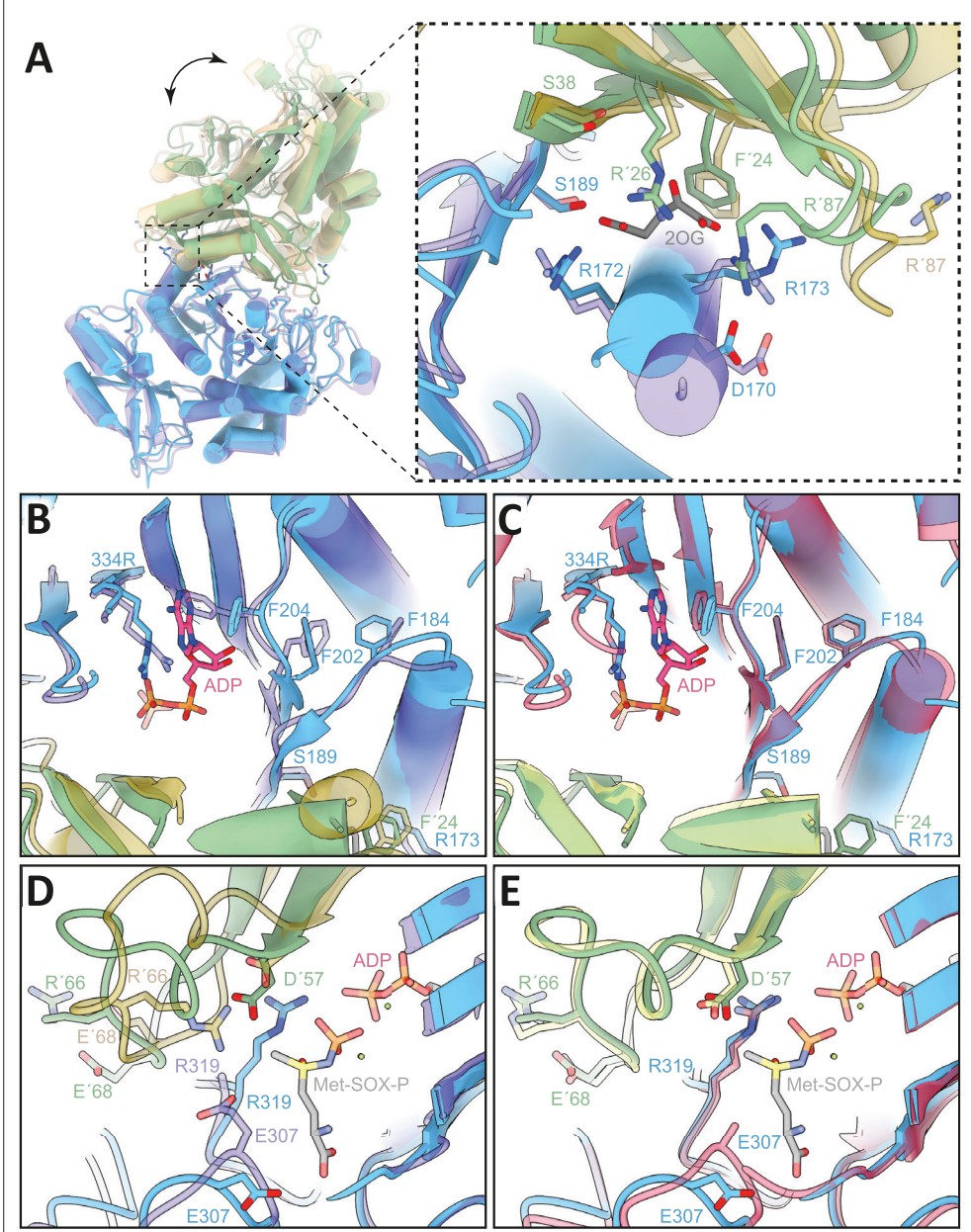

**Figure 5.** Comparison of 2-oxoglutarate (2-OG) and substrate binding site of 2-OG bound, apo, and transition state (TS) structures (.*Schumacher et al., 2023*). Atomic models in cartoon, important residues shown in stick representation. Colors: *M. mazei* GlnA$_1$ 2-OG - blue/green, *M. mazei* GlnA$_1$ apo (PDB: 8tfb, *Schumacher et al., 2023*) - purple/ochre and *M. mazei* GlnA$_1$ Met-Sox-P·ADP (PDB: 8tfk, *Schumacher et al., 2023*) transition state (GlnA$_1$ TS) - red/yellow. (**A**) *left:* GlnA$_1$ 2-OG dimer in superposition with GlnA$_1$ apo showing large-scale movements upon 2-OG binding. (**A**) *right:* Close-up of 2-OG binding site of GlnA$_1$ 2-OG in superposition with GlnA$_1$ apo. Dramatic movement of Helix α3 (residue 167–181) and R87 loop show effect of 2-OG binding. (**B**) Close-up of substrate binding site of GlnA$_1$ 2-OG in superposition with GlnA$_1$ apo and ADP ligand from GlnA$_1$ TS. Helix α3 movement upon 2-OG binding leads to a cascade of conformational changes of the phenylalanines F184, F202, and F204 that lead to a priming of the active site for ATP binding. (**C**) Close-up of substrate binding site of GlnA$_1$ 2-OG in superposition with GlnA$_1$ TS shows high similarity between 2-OG bound and transition state structure. (**D**) Close-up of substrate binding site of GlnA$_1$ 2-OG in superposition with GlnA$_1$ apo and Met-Sox-P ligand from GlnA$_1$ TS. Large structural changes of the D50-loop with ejection of the R66 key-residue are shown. Flipping of the loop allows R319 and D57 to move in further and catalyze phosphoryl-transfer and attack of NH$_4^+$, respectively. (**E**) Close-up of the substrate binding site of GlnA$_1$ 2-OG in superposition with GlnA$_1$ TS reveals a strong similarity between 2-OG bound and transition state structure in the active site.

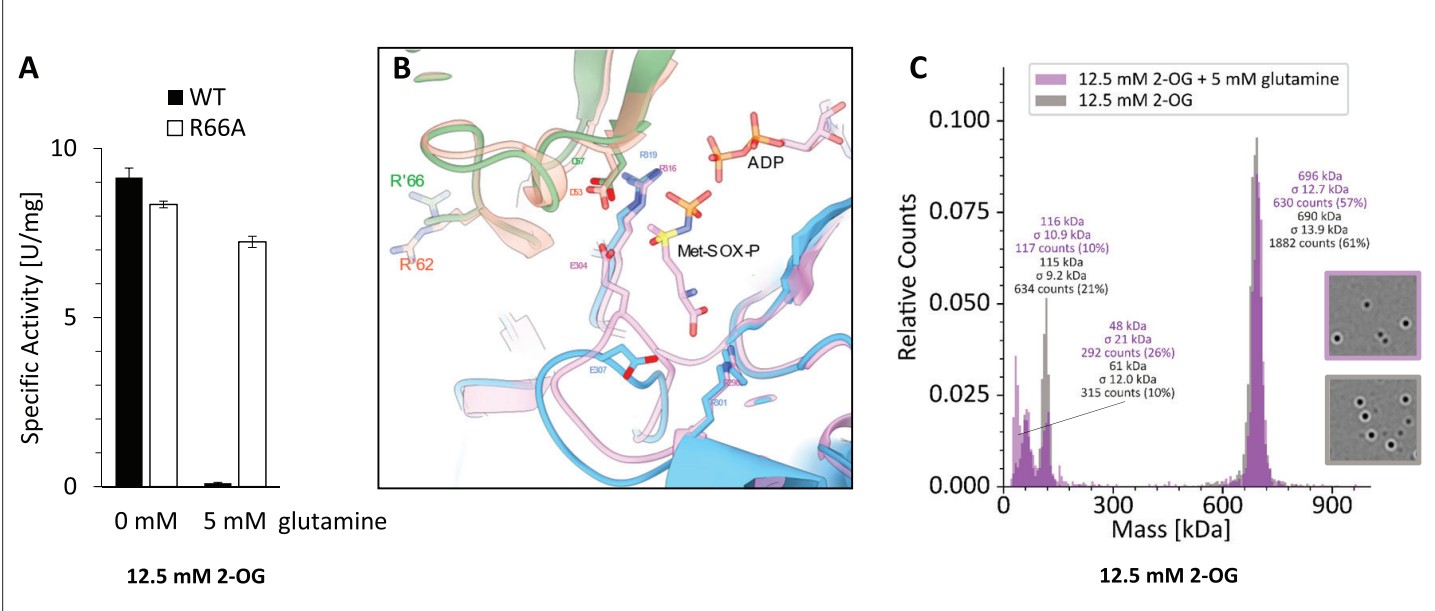

**Figure 6.** Feedback inhibition of GlnA$_1$ by glutamine. (**A**) Specific activity of purified strep-tagged GlnA$_1$ (wt) and the respective R66A-mutant protein was determined as described in Materials and methods in the presence of 12.5 mM 2-oxoglutarate (2-OG) and after additional supplementation of 5 mM glutamine. For wt and the R66A-mutant, one out of two biological independent replicates are exemplarily shown, the deviation indicates the average of four technical replicates. (**B**) Superposition glutamine synthetases (GS) structures without glutamine of *M. mazei* (blue, green) and *B. subtilis* (orange, pink; PDB: 4lnn, *Murray et al., 2013*): substrate binding-site including R'66 (R'62, respectively), which are responsible for feedback inhibition. (**C**) Exemplary mass spectra of Strep-GlnA$_1$ with 12.5 mM 2-OG in presence and absence of 5 mM glutamine. The molecular masses shown above the peaks correspond to a Gaussian fit of the respective peak (Gaussian fit not shown).

exclusively the wt was strongly feedback inhibited, whereas the R66A mutant protein was not significantly affected (*Figure 6A*). In *B. subtilis*, R62 is responsible for feedback inhibition. The superposition of the apo-BsGS structure (PDB: 4lnn, *Murray et al., 2013*) with our 2-OG-bound GlnA$_1$ reveals a similar positioning of the respective *M. mazei* R66 (*Figure 6B*) indicating a similar mechanism. Moreover, we can rule out an effect on the oligomeric structure of GlnA$_1$ by MP analysis, clearly showing that glutamine does not induce disassembly of the dodecameric wt GlnA$_1$ (*Figure 6C*). Instead, this effect can be explained with the role of R66 being an important residue to bind to glutamine in the product state of the enzyme.

## Discussion
### 2-OG is crucially required for *M. mazei* GS assembly to an active dodecamer and induces the conformational shift towards an active open state

In *M. mazei*, increased 2-OG concentrations act as a central N starvation signal (*Ehlers et al., 2005*). Here, we demonstrated the importance of 2-OG as the major regulator of *M. mazei* GlnA$_1$ activity by using independent methods, MP and cryo-EM, to detect and structurally characterize single complexes with high resolution and quantify different oligomeric states of GlnA$_1$. We have found mainly dimeric GlnA$_1$ (apo GlnA$_1$) to be inactive and crucially require 2-OG to form an active dodecameric complex. Moreover, this dodecameric conformation is the only active state of GlnA$_1$. In the first step, 2-OG assembles the dodecamer by binding at the interface of two subunits (*Figure 4B*) and functions as a molecular glue between neighboring subunits. The assembly upon 2-OG addition observed using MP appears to be cooperative, fast, and without any detectable intermediate states (*Figure 1B and C*). Only immediately after thawing a frozen purified GlnA, and in case no additional SEC was performed prior to MP analysis, samples showed additional octameric complexes in MP with low abundancy (*Figure 1—figure supplement 3*). However, octameric complexes were never observed in cryo-EM or detected by SEC analysis of frozen purified GlnA$_1$ samples. Consequently,

octamers are most likely broken or disassembled $GlnA_1$-dodecamers or dead-ends in assembly with no physiological function, rather than an incomplete dodecamer during assembly. Thus, our findings are contrary to the assembly model proposed by *Schumacher et al., 2023*.

As a second step of activation, the allosteric binding of 2-OG causes a series of conformational changes in $GlnA_1$ protomers, which prime the active site for the transition state and hence catalysis of the enzyme. This conformational change of the ATP-binding pocket of the dodecameric $GlnA_1$ upon 2-OG binding goes hand in hand with the observed increased activity at higher 2-OG concentrations (*Figure 1*). Comparing our 2-OG-bound $GlnA_1$ dodecameric structure and the dodecameric *M. mazei* $GlnA_1$ transition state (PDB: 8tfk) and apo structures (PDB: 8ftb) reported by *Schumacher et al., 2023*, clearly demonstrates that 2-OG transfers $GlnA_1$ into its open active state conformation (*Figure 5*). The conformation of our 2-OG-bound dodecamer resembled the transition state conformation (ADP-Met-Sox-bound complex) reported by Schumacher et al., even though in our case no ATP was added (*Figure 5E*). A reconfiguration of the active site upon 2-OG-binding has also been reported for GS in *Methanothermococcus thermolithotrophicus* (*Müller et al., 2024*). In this report, which does not delineate dodecamer assembly at all, it was demonstrated that binding of 2-OG in one protomer-protomer interface of a dodecameric GS causes a cooperative domino effect in the hexameric ring of *M. thermolithotrophicus* GS (*Müller et al., 2024*). A 2-OG bound protomer undergoes a conformational change and thereby induces the same shift in its neighboring protomer (*Müller et al., 2024*). This is comparable to our observed cooperativity of *M. mazei* dodecamer assembly at low 2-OG concentrations (EC50=0.75 mM, calculated based on the percentage of dodecamer). On the other hand, *M. mazei* $GlnA_1$ reaches maximal activity only at much higher 2-OG concentrations (EC50=6.3 mM 2-OG) and likely requires a fully 2-OG-occupied dodecamer for maximal activity. The difference between the two EC50 values strongly points towards the dodecamer assembly being induced by only one 2-OG per hexameric ring, whilst the maximum activity requires one 2-OG in every 2-OG binding site (in agreement with roughly sixfold higher EC50). The here obtained high activities by 2-OG saturation (up to 9 U/mg), in comparison with previously described *M. mazei* $GlnA_1$ activities in the absence of 2-OG in a significantly lower range (mU/mg) (*Gutt et al., 2021*; *Schumacher et al., 2023*), support our conclusion that 2-OG is substantial for the $GlnA_1$ active state.

## $GlnA_1$ activity is further regulated by feedback inhibition, small proteins, and possibly filament formation

*M. mazei* $GlnA_1$ belongs to the group of Iα-type GS, which are known to be feedback inhibited. We confirmed a strong feedback inhibition by a genetic approach and validated R66 to be the key residue for this inhibition (*Figure 6*) as suggested in *Müller et al., 2024*. The mechanism of feedback inhibition has been described in detail for *B. subtilis* GS (*Murray et al., 2013*). There, R62 plays the central role by binding glutamine and inducing a well-ordered inactive structure at the substrate-binding pocket upon glutamine-binding (*Murray et al., 2013*). The homologous *M. mazei* R66 likely conveys a similar way of inhibition to *B. subtilis* GS (*Figure 6B*, *Figure 7—figure supplement 1*).

Further regulations by the two small proteins sP26 and the PII-like protein $GlnK_1$ have previously been reported for *M. mazei* (*Ehlers et al., 2005*; *Gutt et al., 2021*; *Schumacher et al., 2023*). Moreover, in previous reports, $GlnK_1$ was shown to interact with $GlnA_1$ in vivo after a nitrogen upshift by pull-down approaches (*Ehlers et al., 2005*), pointing towards an inhibitory function of $GlnK_1$ under shifting conditions from N limitation to N sufficiency. However, in the present study, neither an interaction with $GlnK_1$, nor $GlnK_1$ effects on $GlnA_1$ complex formation analyzed by MP, nor an effect of $GlnK_1$ on $GlnA_1$ activity was detectable under the conditions used at varying 2-OG concentrations (0.1–12.5 mM) and ratios of $GlnK_1$ to $GlnA_1$ (20:1, 2:1, 2:10) (*Figure 2*). Moreover, the addition of $GlnK_1$ did not result in a change of the EC50 of 2-OG for the dodecamer $GlnA_1$ assembly (*Figure 2D*). Similarly, we could not determine a cryo-EM structure including sP26 despite adding a large excess of the small protein either obtained by co-expression or by addition of a synthetic peptide. Because these attempts were unsuccessful, we speculate that yet unknown cellular factor(s) might be required for an interaction of $GlnA_1$ with both small proteins, $GlnK_1$ and sP26, which however is difficult to simulate under in vitro conditions with purified proteins. Taking this into account, we speculate about a potential function of the two small proteins beyond $GlnA_1$ inactivation or activation. Since the $GlnA_1$ reaction is coupled to the GOGAT reaction (GS/GOGAT system) and the products of the two reactions replenish the substrates for one other, it is tempting to speculate that $GlnA_1$ and GOGAT experience

metabolic coupling by sP26 and/or GlnK$_1$, e.g., by being involved in recruiting or separating GOGAT from GlnA$_1$.

Finally, higher oligomeric states of GS enzymes have been known for a long time for organisms like yeast and *E. coli* (*He et al., 2009*; *Huang et al., 2022*; *Petrovska et al., 2014*; *Valentine et al., 1968*). Interestingly, dependent on the ice thickness and on higher concentrated areas of the grids, we could also observe filament-like structures of *M. mazei* GlnA$_1$ in cryo-EM and resolved their structure at a resolution of 6.9 Å (*Figure 3—figure supplement 2*). Such GlnA$_1$ filaments are also detectable in the cryo-EM images of *Schumacher et al., 2023*, but were not reported. The filament interface is much alike the previously reported *E. coli* GS filament structures (*Huang et al., 2022*). The physiological relevance of filamentation in *M. mazei* however remains unresolved and raises the question, whether an additional rapid modulation of GlnA$_1$ activity through higher oligomeric states exists. In yeast for example, GS filamentation was described as mostly depending on stress conditions (*Petrovska et al., 2014*).

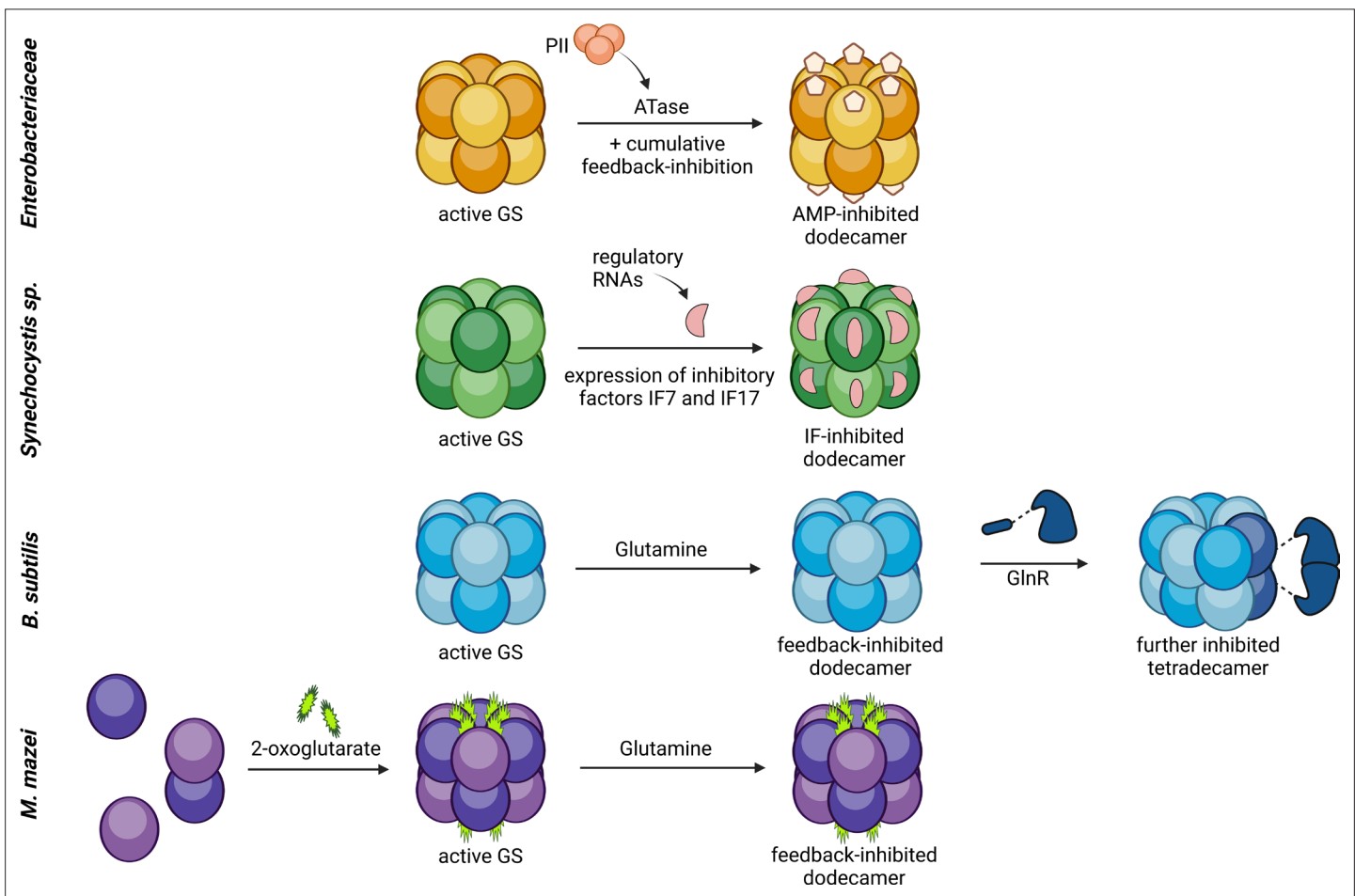

**Figure 7.** Model of the various molecular mechanisms of glutamine synthetase activity regulation. Comparison of the regulation of glutamine synthetase activity in *E. coli /Salmonella typhimurium*, *B. subtilis*, *Synechocystis*,a and *M. mazei*. Glutamine synthetases (GS) are in general active in a dodecameric, unmodified complex under nitrogen limitation. Upon an ammonium upshift, GS are inactivated by feedback inhibition (BcGS, *E. coli*), covalent modification (adenylylation, EcGS) or binding of (small) inactivating proteins (*Synechocystis*, BsGS). *M. mazei* GS on the contrary is regulated via the assembly of the active dodecamer upon 2-oxoglutarate (2-OG)-binding and furthermore is strongly feedback inhibited by glutamine. (*Bolay et al., 2018*; *Klähn et al., 2018*; *Klähn et al., 2015*; *Stadtman, 2001*; *Travis et al., 2022*). Created with BioRender.com.

The online version of this article includes the following figure supplement(s) for figure 7:

**Figure supplement 1.** Amino acid sequence alignment of different model organism glutamine synthetases.

### M. mazei GlnA$_1$ shows unique properties

Overall, we have confirmed 2-OG to be the central activator of GlnA$_1$ in *M. mazei,* which assembles the active dodecamer and induces a conformational switch towards an active open state. Though 2-OG has previously been reported as an on-switch for (methano)archaeal GS activity (*Ehlers et al., 2005*; *Müller et al., 2024*; *Pedro-Roig et al., 2013*), the 2-OG-triggered dodecameric assembly is novel and described exclusively for *M. mazei* GlnA$_1$. Neither in cyanobacteria, enterobacteria or *Bacillus* has 2-OG been reported as the sole direct activator of the enzyme, nor is complex (dis-) assembly a mode of regulating GS activity in any other of these model organisms. This is further supported by the absence of up to three of those four arginines, which are coordinating 2-OG in *M. mazei* GlnA$_1$, in these organisms (*Figure 7—figure supplement 1*). The cyanobacterial, enterobacterial, and gram-positive GS are assumed to be present in the cell as active dodecamers (*Almassy et al., 1986*; *Bolay et al., 2018*; *Deuel et al., 1970*). These dodecamers are inactivated upon sudden N sufficiency through very different mechanisms all including additional proteins (see *Figure 7*). In *Synechocystis,* GS is blocked by small proteins, which are repressed under nitrogen limitation, one in a 2-OG-NtcA mediated way and the other one via a glutamine sensing riboswitch (*Bolay et al., 2018*; *Klähn et al., 2018*; *Klähn et al., 2015*). The enterobacterial GS experiences 2-OG-PII dependent gradual adenylylation of subunits, which abolishes the enzyme activity, and *B. subtilis* GS is feedback inhibited by glutamine and further inhibited by binding of the transcription factor GlnR (*Almassy et al., 1986*; *Stadtman, 2001*; *Travis et al., 2022*). Consequently, the GS regulation in *M. mazei* by a 2-OG triggered assembly is unique across all prokaryotic GS studied so far.

The direct 2-OG activation and glutamine feedback inhibition of *M. mazei* GS are two fast, reversible, and very direct ways of reacting towards the changing N status of the cell. We propose that the direct activation through 2-OG without any required additional protein, as it is the case for all other regulations, is a more simple and direct regulation of GS. Due to the evolutionary placement of methanoarchaea and haloarchaea, where a direct 2-OG regulation has been found exclusively, this may represent an ancient regulation.

## Materials and methods
### Strains and plasmids

For heterologous expression and purification of Strep-tagged GlnA$_1$ (MM_0964), the plasmid pRS1841 was constructed. The glnA$_1$-sequence along with the sP26-sequence (including start-codon: ATG) were codon-optimized for *Escherichia coli* expression and commercially synthesized by Eurofins Genomics on the same plasmid (pRS1728) (Ebersberg, Germany). Polymerase chain reaction (PCR) was performed using pRS1728 as template and the primers (Eurofins Genomics, Ebersberg, Germany) GlnAopt_NdeI_for (5'TTTCATATGGTTCAGATGAAAAAATG3') and GlnA1opt_BamHI_rev (5'TTTG GATCCTTACAGCATGCTCAGATAACGG3'). The resulting GlnA$_1$_opt PCR-product and vector pRS375 were restricted with NdeI and BamHI (NEB, Schwalbach, Germany); the resulting pRS375 vector fragment and the GlnA$_1$ fragment were ligated resulting in pRS1841. For heterologous expression of Strep-GlnA$_1$, pRS1841 was transformed in *E. coli* BL21 (DE3) cells (Thermo Fisher Scientific, Waltham, Massachusetts) following the method of Inoue (*Inoue et al., 1990*). For generating the Arg66Ala-mutant, a site-directed mutagenesis was performed. pRS1841 was PCR-amplified using primers sdm_GlnA_R66A_for (5'ATTGAAGAAAGCGATATGAAACTGGCGC3') and sdm_GlnA_R66A_rev (5'**CGC**GGTAAAGCCCTGAATGCTGCTACC3') by Phusion High-Fidelity polymerase (Thermo Fisher Scientific, Waltham, Massachusetts) followed by religation resulting in plasmid pRS1951. For heterologous expression, pRS1951 was transformed into *E. coli* BL21 (DE3).

In order to co-express sP26 along with Strep-GlnA$_1$, the construct pRS1863 was generated. pRS1728 with the codon-optimized sP26-sequence and pET21a (Novagen, Darmstadt, Germany) were restricted with NdeI and NotI and the resulting untagged sP26_opt was ligated into the pET21a backbone yielding pRS1863. pRS1863 was co-transformed with pRS1841 into *E. coli* BL21 (DE3) cells (Thermo Fisher Scientific, Waltham, Massachusetts) selected for both Kanamycin (pRS1841 derived) and Ampicillin (pRS1863 derived) resistance.

The plasmid pRS1672 was constructed for producing untagged GlnK$_1$. The GlnK$_1$ gene was PCR-amplified using primers GlnK1_MM0732.for (5'ATGGTTGGCTATGAAATACGTAATTG3') and GlnK1_MM0732.rev (5'TCAAATTGCCTCAGGTCCG3') and cloned into pETSUMO by using the Champion

pET SUMO Expression System (Thermo Fisher Scientific, Waltham, Massachusetts) according to the manufacturer's protocol. pRS1672 was then transformed into *E. coli* DH5α (*Hanahan, 1983*) and BL21 (DE3) pRIL (*Supplementary file 1*).

## Heterologous expression and protein purification: Strep-GlnA$_1$ and GlnK$_1$

Heterologous expression of Strep-GlnA$_1$-variants (pRS1841 and pRS1951) and Strep-GlnA$_1$-sP26-coexpression (pRS1841 + pRS1863) were performed in 1 l Luria Bertani medium (LB, Carl Roth GmbH + Co. KG, Karlsruhe, Germany). *E. coli* BL21 (DE3) containing pRS1841, pRS1841 and pRS1863 or pRS1951 was grown to an optical turbidity at 600 nm ($T_{600}$) of 0.6–0.8, induced with 25 µM isopropylβ-d-1-thiogalactopyranoside (IPTG, Carl Roth GmbH + Co. KG, Karlsruhe, Germany) and further incubated overnight at 18 °C and 120 rpm. The cells were harvested (6371 × g, 20 min, 4 °C) and resuspended in 6 ml W-buffer (100 mM TRIS/HCl, 150 mM NaCl, 2.5 mM EDTA, (chemicals from Carl Roth GmbH + Co. KG, Karlsruhe, Germany), 12.5 mM 2-oxoglutarate (2-OG, Sigma-Aldrich, St. Louis, Missouri), pH 8.0). After the addition of DNase I (Sigma-Aldrich, St. Louis, Missouri), cell disruption was performed twice using a French Pressure Cell at $4.135 \times 10^6$ N/m$^2$ (*Sim-Aminco Spectronic Instruments*, Dallas, Texas) followed by centrifugation of the cell lysate for 30 min (13,804 × g, 4 °C). The supernatant was incubated with 1 ml equilibrated (W-buffer) Strep-Tactin sepharose matrix (IBA, Gottingen, Germany) at 4 °C for 1 h at 20 rpm. Strep-tagged GlnA$_1$ was eluted from the gravity flow column by adding an E-buffer (W-buffer +2.5 mM desthiobiotine (IBA, Gottingen, Germany)). Strep-GlnA$_1$ was always purified and stored in the presence of 12.5 mM 2-OG, either in E-buffer or 50 mM HEPES, pH 7.0 at 4 °C for a few days or with 5% glycerol at –80 °C (chemicals from Carl Roth GmbH + Co. KG, Karlsruhe, Germany).

His$_6$-SUMO-GlnK$_1$ was expressed similarly using *E. coli* BL21 (DE3) pRIL +pRS1672. Expression was induced with 100 µM IPTG, incubated at 37 °C, 180 rpm for 2 hr, and harvested. The pellet was resuspended in phosphate buffer (50 mM phosphate, 300 mM NaCl, pH 8 (chemicals from Carl Roth GmbH + Co. KG, Karlsruhe, Germany)) and the cell extract was prepared as described above. His-tag-affinity chromatography-purification was performed with a Ni-NTA agarose (Qiagen, Hilden, Germany) gravity flow column, the protein was purified by stepwise-elution with 100 and 250 mM imidazole (SERVA, Heidelberg, Deutschland) in phosphate buffer. SUMO-protease (Thermo Fisher Scientific, Waltham, Massachusetts) was used according to the manufacturer's protocol to cleave the His$_6$-SUMO-GlnK$_1$ and obtain untagged GlnK$_1$ by passing through the Ni-NTA-column after the cleavage. Elution fractions of protein purifications were analyzed on 12% SDS-PAGE gels and the protein concentrations were determined by Bradford (Bio-Rad Laboratories, Hercules, California) or Qubit protein assay (Thermo Fisher Scientific, Waltham, Massachusetts).

## Determination of glutamine synthetase activity

The glutamine synthetase activity was determined by performing a coupled optical assay (*Shapiro and Stadtman, 1970*). The assay was performed as described in *Gutt et al., 2021* with modifications. First, a substrate mix containing 257 mM KCl, 143 mM NH$_4$Cl, 143 mM MgCl$_2$ (chemicals from Carl Roth GmbH + Co. KG, Karlsruhe, Germany), and 86 mM sodium-glutamate (Sigma-Aldrich, St. Louis, Missouri) was prepared. The assay was performed in a final volume of 1 ml including 350 µl of the substrate mix, 10 µl of lactic dehydrogenase and pyruvate kinase mix (Sigma-Aldrich, St. Louis, Missouri), 50 mM HEPES (final concentration, Carl Roth GmbH +Co. KG, Karlsruhe, Germany), the respective amount of 2-OG (Sigma-Aldrich, St. Louis, Missouri), 1 mM phosphoenolpyruvate (PEP, Sigma-Aldrich, St. Louis, Missouri), 0.42 mM nicotinamide adenine dinucleotide (NADH) and 10 or 20 µg of Strep-GlnA$_1$. After preincubation at room temperature in a volume of 950 µl for 5 min, the assay mixture was transferred to a cuvette, the time course measurement at 340 nm was started and the enzyme reaction induced by adding 3.6 mM ATP (pH adjusted to 7.0, Roche, Basel, Switzerland) (*Figure 1—figure supplement 4*). Biological replicates were independent protein expressions and purifications and the assays were performed with four technical replicates per condition, including two concentrations of GnA$_1$ (2x10 µg and 2×20 µg of Strep-GlnA$_1$, present in 100 µl were added). Strep-GlnA$_1$ was stored in E-buffer (described above) or 50 mM HEPES containing 12.5 mM 2-OG which was dialyzed against 50 mM HEPES pH 7.0 using Amicon Ultra cartridges with 30 kDa filters (MilliporeSigma, Burlington, Massachusetts) for the enzyme assays in the absence of 2-OG.

## Mass photometry

The molecular weight of protein complexes was analyzed by mass photometry (MP) using a Refeyn twoMP mass photometer with the AcquireMP software (Refeyn Ltd., Oxford, UK). All measurements were performed in 50 mM HEPES, 150 mM NaCl pH 7.0 (MP-buffer, chemicals from Carl Roth GmbH + Co. KG, Karlsruhe, Germany) on 1.5 H, 24×60 mm microscope coverslips with Culture Well Reusable Gaskets (GRACE BIO-LABS, Bend, Oregon). Strep-GlnA$_1$ and untagged GlnK$_1$ were prepared as described above. Prior to MP experiments, a size exclusion chromatography (SEC) was performed with GlnA$_1$ in the presence of 12.5 mM 2-OG on a Superose 6 Increase 10/300 GL column (Cytiva, Marlborough, Massachusetts) with a flow rate of 0.5 ml/min. Only the dodecameric fraction was used for MP experiments and dialyzed against MP buffer using Amicon Ultra cartridges with 30 kDa filters (MilliporeSigma, Burlington, Massachusetts) beforehand. The Gel Filtration HMW Calibration Kit (Cytiva, Marlborough, Massachusetts) was used as a standard in SEC. 75–200 nM monomeric Strep-GlnA$_1$ were used in the MP measurements, GlnK$_1$ was added accordingly in the desired ratio calculated based on monomers. Biological replicates were independent protein expressions and purifications and technical replicates were the repetition of the measurement with the same protein in a fresh mix. The analysis of the acquired data was performed with the DiscoverMP software by applying a premeasured standard (Refeyn Ltd., Oxford, UK). Counts were visualized in mass histograms as relative counts, which were calculated for the Gaussian fits of the measured peaks. For the determination of EC50-values and creating sigmoidal fitted curves, RStudio (RStudio Team (2020). RStudio: Integrated Development for R. RStudio, PBC, Boston, MA URL) was used.

## Cryo-electron sample preparation and data collection

Purified GS at a concentration of 1.5 mg/mL was rapidly applied to glow-discharged Quantifoil grids, blotted with force 4 for 3.5 s, and vitrified by directly plunging in liquid ethane (cooled by liquid nitrogen) using Vitrobot Mark IV (Thermo Fisher Scientific, Waltham, Massachusetts) at 100% humidity and 4 °C. To overcome the preferred orientation bias, 0.7 mM CHAPSO was added to prevent water-air interface interactions, consequently the concentration of the protein was increased to 6 mg/ml. We added purified commercially synthesized sP26 (Davids Biotechnologie, Regensburg, Germany) to all samples, but the peptide did not stably bind under the observed conditions. Data was acquired with EPU in EER-format on an FEI Titan Krios G4 (Cryo-EM Platform, Helmholtz Munich) equipped with a Falcon IVi detector (Thermo Fisher Scientific, Waltham, Massachusetts) with a total electron dose of ~55 electrons per Å$^2$ and a pixel size of 0.76 Å. Micrographs were recorded in a defocus range of –0.25 to –2.0 μm. For details see *Supplementary file 2*.

## Cryo-EM - Image processing, classification, and refinement

All data was processed using Cryosparc (*Punjani et al., 2017*). Micrographs were processed on the fly (motion correction, CTF estimation). Using blob picker, 878,308 particles were picked, 2D-classified, and used for ab initio reconstruction. Iterative rounds of ab initio and heterogenous refinement were used to clean the particle stacks. The final refinements yielded models with an estimated resolution of 2.39 Å sets at the 0.143 cutoff (*Figure 3—figure supplement 1*).

An initial model was generated from the protein sequences using alphaFold (*Jumper et al., 2021*), and thereupon fitted as rigid bodies into the density using UCSF Chimera (*Pettersen et al., 2021*). The model was manually rebuilt using *Coot* (*Emsley et al., 2010*). The final model was subjected to real-space refinements in PHENIX (*Liebschner et al., 2019*). Illustrations of the models were prepared using UCSF ChimeraX (*Pettersen et al., 2021*). The structure is accessible under PDB: 8s59. For details see *Supplementary file 2*.

## Acknowledgements

We thank the members of our laboratories for useful discussions on the experiments, as well as Claudia Kiessling for technical assistance. This work was supported by the German Research Council (DFG) priority program (SPP) 2002 'Small proteins in Prokaryotes, an unexplored world' [Schm1052/20-2]. We acknowledge the contribution of the CryoEM Facility of the Philipps University of Marburg. JMS acknowledges the DFG for an Emmy Noether grant (SCHU 3364/1–1) co-funded by the European Union (ERC, TwoCO2One, 101075992). Views and opinions expressed are, however, those of the

author(s) only and do not necessarily reflect those of the European Union or the European Research Council. Neither the European Union nor the granting authority can be held responsible for them. We thank Sandra Schuller for useful discussions and help in preparing manuscript figures. GKAH was supported by the Max Planck Society. We acknowledge financial support by Land Schleswig-Holstein within the funding program Open Access Publikationsfonds.

## Additional information

### Funding

| Funder | Grant reference number | Author |
|--------|------------------------|--------|
| Deutsche Forschungsgemeinschaft | Schm1052/20-2 | Ruth Anne Schmitz |
| Deutsche Forschungsgemeinschaft | SCHU 3364/1-1 | Jan Schuller |
| European Research Council | 10.3030/101075992 | Jan Schuller |

The funders had no role in study design, data collection and interpretation, or the decision to submit the work for publication.

### Author contributions

Eva Herdering, Tristan Reif-Trauttmansdorff, Investigation, Visualization, Writing – original draft, Writing – review and editing; Anuj Kumar, Data curation, Formal analysis; Tim Habenicht, Stefan Bohn, Investigation; Georg Hochberg, Resources, Supervision, Investigation; Jan Schuller, Conceptualization, Resources, Supervision, Methodology, Writing – original draft, Writing – review and editing; Ruth Anne Schmitz, Conceptualization, Resources, Supervision, Funding acquisition, Methodology, Writing – original draft, Writing – review and editing

### Author ORCIDs

Ruth Anne Schmitz https://orcid.org/0000-0002-6788-0829

Reviewer #1 (Public review): https://doi.org/10.7554/eLife.97484.3.sa1
Reviewer #2 (Public review): https://doi.org/10.7554/eLife.97484.3.sa2
Reviewer #3 (Public review): https://doi.org/10.7554/eLife.97484.3.sa3
Author response https://doi.org/10.7554/eLife.97484.3.sa4

## Additional files

### Supplementary files

Supplementary file 1. Strains and plasmids.

Supplementary file 2. Cryo-electron microscopy (Cryo-EM) data collection, refinement, and validation statistics.

MDAR checklist

### Data availability

Diffraction data have been deposited in PDB under the accession code 8S59.

The following dataset was generated:

| Author(s) | Year | Dataset title | Dataset URL | Database and Identifier |
|-----------|------|---------------|-------------|-------------------------|
| Kumar A, Schuller JM, Schmitz RA | 2025 | Cryo-EM structure of the active dodecameric Methanosarcina mazei glutamine synthetase | https://www.rcsb.org/structure/8S59 | RCSB Protein Data Bank, 8S59 |

The following previously published datasets were used:

| Author(s) | Year | Dataset title | Dataset URL | Database and Identifier |
|---|---|---|---|---|
| Murray DS, Chinnam N, Tonthat NK, Whitfill T, Wray LV, Fisher SH, Schumacher MA | 2013 | *B. subtilis* glutamine synthetase structures reveal large active site conformational changes and basis for isoenzyme specific regulation: structure of apo form of GS | https://doi.org/10.2210/pdb4LNN/pdb | Worldwide Protein Data Bank, 10.2210/pdb4LNN/pdb |
| Schumacher MA, Salinas R, Travis BA, Singh RR, Lent N | 2023 | Cryo-EM structure of the Methanosarcina mazei glutamine synthetase (GS) with Met-Sox-P and ADP | https://doi.org/10.2210/pdb8TFK/pdb | Worldwide Protein Data Bank, 10.2210/pdb8TFK/pdb |
| Schumacher MA, Salinas R, Travis BA, Singh RR, Lent N | 2023 | Cryo-EM structure of the Methanosarcina mazei apo glutamin synthetase structure: dodecameric form | https://doi.org/10.2210/pdb8TFB/pdb | Worldwide Protein Data Bank, 10.2210/pdb8TFB/pdb |

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
