## [Editor Report · eLife Assessment]

This study reveals a novel mechanism of glutamine synthetase (GS) regulation in Methanosarcina mazei, demonstrating that 2-oxoglutarate (2-OG) directly promotes GS activity by stabilizing its dodecameric assembly. Using mass photometry, activity assays, and cryo-electron microscopy, the authors show that GS transitions from a dimeric, inactive form at low 2-OG concentrations to a fully active dodecameric complex at saturating 2-OG levels, highlighting 2-OG as a key effector in C/N sensing. The findings are **valuable**, supported by **solid** data, and provide new insights into archaeal GS regulation, though further clarification of interactions with known partners like Glnk1 and sp26 is needed.

---

## [Referee Report · Reviewer #1 (Public review)]

Summary:

Shows a new mechanism of GS regulation in the archaean Methanosarcina maze and clarifies the direct activation of GS activity by 2-oxoglutarate, thus featuring an other way, how 2-oxoglutarate acts as a central status reporter of C/N sensing.

Strengths:

mass photometry reveals a a dynamic mode the effect of 2-OG on the oligomerization state of GS. Single particle Cryo-EM reveals the mechanism of 2-OG mediated dodecamer formation.

Weaknesses:

Not entirely clear, how very high 2-OG concentrations activate GS beyond dodecamer formation.

In the revised version, most of my concerns were adequately addressed. In the summary it is stated that glutamine acts as allosteric inhibitor of dodecameric GS. This is not correct: glutamine binds to the active site and is therefore not allosteric. This way of feedback inhibition is a type of product inhibition

---

## [Referee Report · Reviewer #2 (Public review)]

Summary:

Herdering et al. introduced research on an archaeal glutamine synthetase (GS) from Methanosarcina mazei, which exhibits sensitivity to the environmental presence of 2-oxoglutarate (2-OG). While previous studies have indicated 2-OG's ability to enhance GS activity, the precise underlying mechanism remains unclear. Initially, the authors utilized biophysical characterization, primarily employing a nanomolar-scale detection method called mass photometry, to explore the molecular assembly of Methanosarcina mazei GS (M. mazei GS) in the absence or presence of 2-OG. Similar to other GS enzymes, the target M. mazei GS forms a stable dodecamer, with two hexameric rings stacked in tail-to-tail interactions. Despite approximately 40% of M. mazei GS existing as monomeric or dimeric entities in the detectable solution, the majority spontaneously assemble into a dodecameric state. Upon mixing 2-OG with M. mazei GS, the population of the dodecameric form increases proportionally with the concentration of 2-OG, indicating that 2-OG either promotes or stabilizes the assembly process. The cryo-electron microscopy (cryo-EM) structure reveals that 2-OG is positioned near the interface of two hexameric rings. At a resolution of 2.39 Å, the cryo-EM map vividly illustrates 2-OG forming hydrogen bonds with two individual GS subunits as well as with solvent water molecules. Moreover, local sidechain reorientation and conformational changes of loops in response to 2-OG further delineate the 2-OG-stabilized assembly of M. mazei GS.

Strengths & Weaknesses:

The investigation studies into the impact of 2-oxoglutarate (2-OG) on the assembly of Methanosarcina mazei glutamine synthetase (M mazei GS). Utilizing cutting-edge mass photometry, the authors scrutinized the population dynamics of GS assembly in response to varying concentrations of 2-OG. Notably, the findings demonstrate a promising and straightforward correlation, revealing that dodecamer formation can be stimulated by 2-OG concentrations of up to 10 mM, although GS assembly never reaches 100% dodecamerization in this study. Furthermore, catalytic activities showed a remarkable enhancement, escalating from 0.0 U/mg to 7.8 U/mg with increasing concentrations of 2-OG, peaking at 12.5 mM. However, an intriguing gap arises between the incomplete dodecameric formation observed at 10 mM 2-OG, as revealed by mass photometry, and the continued increase in activity from 5 mM to 10 mM 2-OG for M mazei GS. This prompts questions regarding the inability of M mazei GS to achieve complete dodecamer formation and the underlying factors that further enhance GS activity within this concentration range of 2-OG.

Moreover, the cryo-electron microscopy (cryo-EM) analysis provides additional support for the biophysical and biochemical characterization, elucidating the precise localization of 2-OG at the interface of two GS subunits within two hexameric rings. The observed correlation between GS assembly facilitated by 2-OG and its catalytic activity is substantiated by structural reorientations at the GS-GS interface, confirming the previously reported phenomenon of "funnel activation" in GS. However, the authors did not present the cryo-EM structure of M. mazei GS in complex with ATP and glutamate in the presence of 2-OG, which could have shed light on the differences in glutamine biosynthesis between previously reported GS enzymes and the 2-OG-bound M. mazei GS.

Furthermore, besides revealing the cryo-EM structure of 2-OG-bound GS, the study also observed the filamentous form of GS, suggesting that filament formation may be a universal stacking mechanism across archaeal and bacterial species. However, efforts to enhance resolution to investigate whether the stacked polymer is induced by 2-OG or other factors such as ions or metabolites were not undertaken by the authors, leaving room for further exploration into the mechanisms underlying filament formation in GS.

Comments on revisions:

My comments have been addressed adequately.

I recognize that determining the structure of the GS complex bound to ATP and/or other ligands would enhance this study by offering a more comprehensive understanding of 2-oxoglutarate-mediated dodecameric assembly and activation. However, I accept the authors' explanation for not including this aspect in the current work.

---

## [Referee Report · Reviewer #3 (Public review)]

The current manuscript investigates the effect of 2-oxoglutarate (2OG) as modulator of glutamine synthetase (GS). To do this, the authors rely of mass photometry, specific activity measurements and single particle cryo-EM data.

From the results, the authors conclude that the GS from Methanosarcina mazei shifts from a dimeric, non-active state under low concentrations of 2OG, to a dodecameric and fully active complex at saturating concentrations of 2OG.

GS is a crucial enzyme in all domains of life. The dodecameric fold of GS is recurrent amongst prokaryotic and archaea organisms but the enzyme activity can be regulated in distinct ways. This is a very interesting work combining protein biochemistry with structural biology.

A novel role for 2OG is presented for this mesophilic methanoarchaeon, as a crucial effector for the enzyme oligomerization and full reactivity.

The conclusions of this paper are mostly well supported by data, but some aspects of this GS regulation and interaction with known partners like Glnk1 and sp26 need to be clarified and extended.

---

## [Author Response]

The following is the authors’ response to the original reviews.

**Reviewer #1 (Public Review):**
Summary:his study shows a new mechanism of GS regulation in the archaean Methanosarcina mazei and clarifies the direct activation of GS activity by 2-oxoglutarate, thus featuring another way in which 2-oxoglutarate acts as a central status reporter of C/N sensing.Mass photometry and single particle cryoEM structure analysis convincingly show the direct regulation of GS activity by 2-OG promoted formation of the dodecameric structure of GS. The previously recognized small proteins GlnK1 and Sp26 seem to play a subordinate role in GS regulation, which is in good agreement with previous data. Although these data are quite clear now, there remains one major open question: how does 2-OG further increase GS activity once the full dodecameric state is achieved (at 5 mM)? This point needs to be reconsidered.Weaknesses:It is not entirely clear, how very high 2-OG concentrations activate GS beyond dodecamer formation.The data presented in this work are in stark contrast to the previously reported structure of M. mazei GS by the Schumacher lab. This is very confusing for the scientific community and requires clarification. The discussion should consider possible reasons for the contradictory results.Importantly, it is puzzling how Schumacher could achieve an apo-structire of dodecameric GS? If 2-OG is necessary for dodecameric formation, this should be discussed. If GlnK1 doesn't form a complex with the dodecameric GS, how could such a complex be resolved there?In addition, the text is in principle clear but could be improved by professional editing. Most obviously there is insufficient comma placement.

We thank Reviewer #1 for the professional evaluation and raising important points. We will address those comments in the updated manuscript and especially improve the discussion in respect to the two points of concern.

(1) How can GlnA1 activity further be stimulated with further increasing 2-OG after the dodecamer is already fully assembled at 5 mM 2-OG.

We assume a two-step requirement for 2-OG, the dodecameric assembly and the priming of the active sites. The assembly step is based on cooperative effects of 2-OG and does not require the presence of 2-OG in all 2-OG-binding pockets: 2-OG-binding to one binding pocket also causes a domino effect of conformational changes in the adjacent 2-OG-unbound subunit, as also described for *Methanothermococcus thermolithotrophicus* GS in Müller et al. 2023. Due to the introduction of these conformational changes, the dodecameric form becomes more favourable even without all 2-OG binding sites being occupied. With higher 2-OG concentrations present (> 5mM), the activity increased further until finally all 2-OG-binding pockets were occupied, resulting in the priming of all active sites (all subunits) and thereby reaching the maximal activity.

(2) The contradictory results with previously published data on the structure of *M. mazei* by Schumacher et al. 2023.

We certainly agree that it is confusing that Schumacher et al. 2023 obtained a dodecameric structure without the addition of 2-OG, which we claim to be essential for the dodecameric form. 2-OG is a cellular metabolite that is naturally present in *E. coli*, the heterologous expression host both groups used. Since our main question focused on analysing the 2-OG effect on GS, we have performed thorough dialysis of the purified protein to remove all 2-OG before performing MP experiments. In the absence of 2-OG we never observed significant enzyme activity and always detected a fast disassembly after incubation on ice. We thus assume that a dodecamer without 2-OG in Schumacher et al. 2023 is an inactive oligomer of a once 2-OG-bound form, stabilized e.g. by the presence of 5 mM MgCl2.

The GlnA1-GlnK1-structure (crystallography) by Schumacher et al. 2023 is in stark contrast to our findings that GlnK1 and GlnA1 do not interact as shown by mass photometry with purified proteins. A possible reason for this discrepancy might be that at the high protein concentrations used in the crystallization assay, complexes are formed based on hydrophobic or ionic protein interactions, which would not form under physiological concentrations.

Reviewer #2 (Public Review):Summary:Herdering et al. introduced research on an archaeal glutamine synthetase (GS) from Methanosarcina mazei, which exhibits sensitivity to the environmental presence of 2-oxoglutarate (2-OG). While previous studies have indicated 2-OG's ability to enhance GS activity, the precise underlying mechanism remains unclear. Initially, the authors utilized biophysical characterization, primarily employing a nanomolar-scale detection method called mass photometry, to explore the molecular assembly of Methanosarcina mazei GS (M. mazei GS) in the absence or presence of 2-OG. Similar to other GS enzymes, the target M. mazei GS forms a stable dodecamer, with two hexameric rings stacked in tail-to-tail interactions. Despite approximately 40% of M. mazei GS existing as monomeric or dimeric entities in the detectable solution, the majority spontaneously assemble into a dodecameric state. Upon mixing 2-OG with M. mazei GS, the population of the dodecameric form increases proportionally with the concentration of 2-OG, indicating that 2-OG either promotes or stabilizes the assembly process. The cryo-electron microscopy (cryo-EM) structure reveals that 2-OG is positioned near the interface of two hexameric rings. At a resolution of 2.39 Å, the cryo-EM map vividly illustrates 2-OG forming hydrogen bonds with two individual GS subunits as well as with solvent water molecules. Moreover, local side-chain reorientation and conformational changes of loops in response to 2-OG further delineate the 2-OG-stabilized assembly of M. mazei GS.Strengths & Weaknesses:The investigation studies the impact of 2-oxoglutarate (2-OG) on the assembly of Methanosarcina mazei glutamine synthetase (M mazei GS). Utilizing cutting-edge mass photometry, the authors scrutinized the population dynamics of GS assembly in response to varying concentrations of 2-OG. Notably, the findings demonstrate a promising and straightforward correlation, revealing that dodecamer formation can be stimulated by 2-OG concentrations of up to 10 mM, although GS assembly never reaches 100% dodecamerization in this study. Furthermore, catalytic activities showed a remarkable enhancement, escalating from 0.0 U/mg to 7.8 U/mg with increasing concentrations of 2-OG, peaking at 12.5 mM. However, an intriguing gap arises between the incomplete dodecameric formation observed at 10 mM 2-OG, as revealed by mass photometry, and the continued increase in activity from 5 mM to 10 mM 2-OG for M mazei GS. This prompts questions regarding the inability of M mazei GS to achieve complete dodecamer formation and the underlying factors that further enhance GS activity within this concentration range of 2-OG.Moreover, the cryo-electron microscopy (cryo-EM) analysis provides additional support for the biophysical and biochemical characterization, elucidating the precise localization of 2-OG at the interface of two GS subunits within two hexameric rings. The observed correlation between GS assembly facilitated by 2-OG and its catalytic activity is substantiated by structural reorientations at the GS-GS interface, confirming the previously reported phenomenon of "funnel activation" in GS. However, the authors did not present the cryo-EM structure of M. mazei GS in complex with ATP and glutamate in the presence of 2-OG, which could have shed light on the differences in glutamine biosynthesis between previously reported GS enzymes and the 2-OG-bound M. mazei GS.Furthermore, besides revealing the cryo-EM structure of 2-OG-bound GS, the study also observed the filamentous form of GS, suggesting that filament formation may be a universal stacking mechanism across archaeal and bacterial species. However, efforts to enhance resolution to investigate whether the stacked polymer is induced by 2-OG or other factors such as ions or metabolites were not undertaken by the authors, leaving room for further exploration into the mechanisms underlying filament formation in GS.

We thank Reviewer #2 for the detailed assessment and valuable input. We will address those comments in the updated manuscript and clarify the message.

(1) The discrepancy of the dodecamer formation (max. at 5 mM 2-OG) and the enzyme activity (max. at 12.5 mM 2-OG). We assume that there are two effects caused by 2-OG: 1. cooperativity of binding (less 2-OG needed to facilitate dodecamer formation) and 2. priming of each active site. See also Reviewer #1 R.(1). We assume this is the reason why the activity of dodecameric GlnA1 can be further enhanced by increased 2-OG concentration until all catalytic sites are primed.

(2) The lack of the structure of a 2-OG and ATP-bound GlnA1. Although we strongly agree that this would be a highly interesting structure, it seems out of the scope of a typical revision to request new cryo-EM structures. We evaluate the findings of our present study concerning the 2-OG effects as important insights into the strongly discussed field of glutamine synthetase regulation, even without the requested additional structures.

(3) The observed GlnA1-filaments are an interesting finding. We certainly agree with the referee on that point, that the stacked polymers are potentially induced by 2-OG or ions. However, it is out of the main focus of this manuscript to further explore those filaments. Nevertheless, this observation could serve as an interesting starting point for future experiments.

**Reviewer #3 (Public Review)**:Summary:The current manuscript investigates the effect of 2-oxoglutarate and the Glk1 protein as modulators of the enzymatic reactivity of glutamine synthetase. To do this, the authors rely on mass photometry, specific activity measurements, and single-particle cryo-EM data.From the results obtained, the authors convey that glutamine synthetase from Methanosarcina mazei exists in a non-active monomeric/dimeric form under low concentrations of 2-oxoglutarate, and its oligomerization into a dodecameric complex is triggered by higher concentration of 2-oxoglutarate, also resulting in the enhancement of the enzyme activity.Strengths:Glutamine synthetase is a crucial enzyme in all domains of life. The dodecameric fold of GS is recurrent amongst prokaryotic and archaea organisms, while the enzyme activity can be regulated in distinct ways. This is a very interesting work combining protein biochemistry with structural biology.The role of 2-OG is here highlighted as a crucial effector for enzyme oligomerization and full reactivity.Weaknesses:Various opportunities to enhance the current state-of-the-art were missed. In particular, omissions of the ligand-bound state of GnK1 leave unexplained the lack of its interaction with GS (in contradiction with previous results from the authors). A finer dissection of the effect and role of 2-oxoglurate are missing and important questions remain unanswered (e.g. are dimers relevant during early stages of the interaction or why previous GS dodecameric structures do not show 2-oxoglutarate).

We thank Reviewer #3 for the expert evaluation and inspiring criticism.

(1) Encouragement to examine ligand-bound states of GlnK1. We agree and plan to perform the suggested experiments exploring the conditions under which GlnA1 and GlnK1 might interact. We will perform the MP experiments in the presence of ATP. In GlnA1 activity test assays when evaluating the presence/effects of GlnK1 on GlnA1 activity, however, ATP was always present in high concentrations and still we did not observe a significant effect of GlnK1 on the GlnA1 activity.

(2) The exact role of 2-OG could have been dissected much better. We agree on that point and will improve the clarity of the manuscript. See also Reviewer #1 R.1.

(3) The lack of studies on dimers. This is actually an interesting point, which we did not consider during writing the manuscript. Now, re-analysing all our MP data in this respect, GlnA1 is likely a dimer as smallest species. Consequently, we will add more supplementary data which supports this observation and change the text accordingly.

(4) Previous studies and structures did not show the 2-OG. We assume that for other structures, no additional 2-OG was added, and the groups did not specifically analyse for this metabolite either. All methanoarchaea perform methanogenesis and contain the oxidative part of the TCA cycle exclusively for the generation of glutamate (anabolism) but not a closed TCA cycle enabling them to use internal 2-OG concentration as internal signal for nitrogen availability. In the case of bacterial GS from organisms with a closed TCA cycle used for energy metabolism (oxidation of acetyl CoA) like e.g. *E. coli*, the formation of an active dodecameric GS form underlies another mechanism independent of 2-OG. In case of the recent *M. mazei* GS structures published by Schumacher et al. 2023, the dodecameric structure is probably a result from the heterologous expression and purification from *E. coli*. (See also Reviewer #1 R.2). One example of methanoarchaeal glutamine synthetases that do in fact contain the 2-OG in the structure, is Müller et al. 2023.

**Recommendations for the authors:**

**Reviewer #1 (Recommendations For The Authors):**
Specific issues:L 141: 2-OG levels increase due to slowing GOGAT reaction (due to Gln limitation as a consequence of N-starvation).... (2-OG also increases in bacteria that lack GDH...)As the GS-GOGAT cycle is the major route of ammonium assimilation, consumption of 2-OG by GDH is probably only relevant under high ammonium concentrations.

In Methanoarchaea, GS is strictly regulated and expression strongly repressed under nitrogen sufficiency - thus glutamate for anabolism is mainly generated by GDH under N sufficiency consuming 2-OG delivered by the oxidative part of the TCA cycle (Methanogenesis is the energy metabolism in methanoarchaea, a closed TCA cycle is not present) thus 2-OG is increasing under nitrogen limitation, when no NH3 is available for GDH.

L148: it is not clear what is meant by: "and due to the indirect GS activity assay"

We apologize for not being clear here. The GS activity assay used is the classical assay by Shapiro & Stadtman 1970 and is a coupled optical test assay (coupling the ATP consumption of the GS activity to the oxidation of NADH by lactate dehydrogenase). Based on the coupled test assay the measurements of low activities show a high deviation. We now added this information in the revised MS respectively.

L: 177: arguing about 2-OG affinities: more precisely, the 0.75 mM 2-OG is the EC50 concentration of 2-OG for triggering dodecameric formation; it might not directly reflect the total 2-OG affinity, since the affinity may be modulated by (anti)cooperative effects, or by additional sites... as there may be different 2-OG binding sites involved... (same in line 201)

Thank you for the valuable input. We changed KD to EC50 within the entire manuscript. Concerning possible additional 2-OG binding sites: we did not see any other 2-OG in the cryo-EM structure aside from the described one and we therefore assume that the one described in the manuscript is the main and only one. Considering the high amounts of 2-OG (12.5 mM) used in the structure, it is quite unlikely that additional 2-OG sites exist since they would have unphysiologically low affinities.

In this respect, instead of the rather poor assay shown in Figure 1D, a more detailed determination of catalytic activation by different 2-OG concentrations should be done (similar to 1A)... This would allow a direct comparison between dodecamerization and enzymatic activation.

We agree and performed the respective experiments, which are now presented in revised Fig. 1D

Discussion: the role of 2-OG as a direct activator, comparison with other prokaryotic GS: in other cases, 2-OG affects GS indirectly by being sensed by PII proteins or other 2-OG sensing mechanisms (like 2OG-NtcA-mediated repression of IF factors in cyanobacteria)

We agree and have added that information in the discussion as suggested.

290. Unclear: As a second step of activation, the allosteric binding of 2-OG causes a series of conformational.... where is this site located? According to the catalytic effects (compare 1A and 1D) this site should have a lower affinity …

Thank you very much for pointing this out. Binding of 2-OG only occurs in one specific allosteric binding-site. Binding however, has two effects on the GlnA1: dodecamer assembly and priming of the active site (with two specific EC50, which are now shown in Fig. 1A and D).

See also public comment #1 (1).

**Reviewer #2 (Recommendations For The Authors):**
The primary concern for me is that mass photometry might lead to incorrect conclusions. The differences in the forms of GS seen in SEC and MP suggest that GS can indeed form a stable dodecamer when the concentration of GS is high enough, as shown in Figure S1B. I strongly suggest using an additional biophysical method to explore the connection between GS and 2-OG in terms of both assembly and activity, to truly understand 2-OG's role in the process of assembly and catalysis.

We apologize if we did not present this clear enough, however the MP analysis of GlnA1 in the absence of 2-OG showed always (monomers/) dimers, dodecamers were only present in the presence of 2-OG. The SEC analysis in (Fig. 1 - Fig. Suppl. 2B). has been performed in the presence of 12.5 mM 2-OG, we realized this information is missing in the figure legend - we now added this in the revised version. The 2-OG is in addition visible in the Cryo EM structure. Thus, we do not agree to perform additional biophysical methods.

As for the other experimental findings, they appear satisfactory to me, and I have no reservations regarding the cryoEM data.(1) Mass photometry is a fancy technique that uses only a tiny amount of protein to study how they come together. However, the concentration of the protein used in the experiment might be lower than what's needed for them to stick together properly. So, the authors saw a lot of single proteins or pairs instead of bigger groups. They showed in Figure S1B that the M. mazei GS came out earlier than a 440-kDa reference protein, indicating it's actually a dodecamer. But when they looked at the dodecamer fraction using mass photometry, they found smaller bits, suggesting the GS was breaking apart because the concentration used was too low. To fix this, they could try using a technique called analytic ultracentrifuge (AUC) with different amounts of 2-OG to see if they can spot single proteins or pairs when they use a bit more GS. They could also try another technique called SEC-MALS to do similar tests. If they do this, they could replace Figure 1A with new data showing fully formed GS dodecamers when they use the right amount of 2-OG.

Thank you for this input. In MP we looked at dodecamer formation after removing the 2-OG entirely and re-adding it in the respective concentration. We think that GlnA1 is much more unstable in its monomeric/dimeric fraction and that the complete and harsh removal of 2-OG results in some dysfunctional protein which does not recover the dodecameric conformation after dialysis and re-addition of 2-OG. Looking at the dodecamer-peak right after SEC however, we exclusively see dodecamers, which is now included as an additional supplementary figure (Fig. 1 - Fig. Suppl. 2C). Consequently, we did not perform additional experiments.

(2) Building on the last point, the estimated binding strength (Kd) between 2-OG and GS might be lower than it really is, because the GS often breaks apart from its dodecameric form in this experiment, even though 2-OG helps keep the pairs together, as seen with cryoEM. What if they used 5-10 times more GS in the mass photometry experiment? Would the estimated bond strength stay the same? Could they use AUC or other techniques like ITC to find out the real, not just estimated, strength of the bond?

We agree that the term KD is not suitable. We have changed the term KD to EC50 as suggested by reviewer #1, which describes the effective concentration required for 50 % dodecamer assembly. Furthermore, we disagree that the dodecamer breaks apart when the concentrations are as low as in MP experiments. The actual reason for the breaking is rather the harsh dialysis to remove all 2-OG before MP experiments. Right after SEC, the we exclusively see dodecamer in MP (Fig. 1 - Fig. Suppl. 2C). See also #2 (1).

(3) The fact that the GS hardly works without 2-OG is interesting. I tried to understand the experiment setup, but it wasn't clear as the protocol mentioned in the author's 2021 FEBS paper referred to an old paper from 1970. The "coupled optical test assay" they talked about wasn't explained well. I found other papers that used phosphometry assays to see how much ATP was used up. I suggest the authors give a better, more detailed explanation of their experiments in the methods section. Also, it's unclear why the GS activity keeps going up from 5 to 12.5 mM 2-OG, even though they said it's saturated. They suggested there might be another change happening from 5 to 12.5 mM 2-OG. If that's the case, they should try to get a cryo-EM picture of the GS with lots of 2-OG, both with and without ATP/glutamate (or the Met-Sox-P-ADP inhibitor), to see what's happening at a structural level during this change caused by 2-OG.

We agree with the reviewer that the GS assay was not explained in detail (since published and known for several years). However, we now added the more detailed description of the assay in the revised MS, which also measures the ATP used up by GS, but couples the generation of ADP to an optical test assay producing pyruvate from PEP with the generated ADP catalysed by pyruvate kinase present in the assay. This generated pyruvate is finally reduced to lactate by the present lactate dehydrogenase consuming NADH, the reduction of which is monitored at 340 nm.

The still increasing activity of GS after dodecamer formation (max. at 5 mM 2-OG) and the continuously increasing enzyme activity (max. at 12.5 mM 2-OG): See also public reviews, we assume that there are two effects caused by 2-OG: 1. cooperativity of binding (less 2-OG needed to facilitate dodecamer formation) and 2. priming of each active site.

The suggested additional experiments with and without ATP/Glutamate: Although we strongly agree that this would be a highly interesting structure, it seems out of the scope of a typical revision to request new cryo-EM structures. We evaluate the findings of our present study concerning the 2-OG effects as important insights into the strongly discussed field of glutamine synthetase regulation, even without the requested additional structures.

(4) Please remake Figure S2, the panels are too small to read the words. At least I have difficulty doing so.

We assume the reviewer is pointing to Fig. 3 - Fig. Suppl. 1, we now changed this figure accordingly.

Line 153, the reference Schumacher et al. 23, should be 2023?

Yes, thank you. We corrected that.

Line 497. I believe it's UCSF ChimeraX, not Chimera.

We apologize and corrected accordingly.

**Reviewer #3 (Recommendations For The Authors):**
Recent studies on the Methanothermococcus thermolithotrophicus glutamine synthetase, published by Müller et al., 2024, have identified the binding site for 2-oxoglutarate as well as the conformational changes that were induced in the protein by its presence. In the present study, the authors confirm these observations and additionally establish a link between the presence of 2-oxoglutarate and the dodecameric fold and full activation of GS.Curiously, here, the authors could not confirm their own findings that the dodecameric GS can directly interact with the PII-like GlnK1 protein and the small peptide sP26. However, the lack of mention of the GlnK-bound state in these studies is very alarming since it certainly is highly relevant here.

We agree with the reviewer that we have not observed the interaction with GlnK1 and sP26 in the recent study. Consequently, we speculate that yet unknown cellular factor(s) might be required for an interaction of GlnA1 with GlnK1 and sP26, which were not present in the in vitro experiments using purified proteins, however they were present in the previous pull-down approaches (Ehlers et al. 2005, Gutt et al. 2021). Another reason might be that post-translational modifications occur in *M. mazei*, which might be important for the interaction, which are also not present in purified proteins expressed in *E. coli*.

The manuscript interest could have been substantially increased if the authors had done finer biochemical and enzymatic analyses on the oligomerization process of GS, used GlnK1 bound to known effectors in their assays and would have done some more efforts to extrapolate their findings (even if a small niche) of related glutamine synthetases.

We thank the reviewer for their valuable encouragement to explore ligand-bound-states of GlnK1. However, in this manuscript we mainly focused on 2-OG as activator of GlnA1 and decided to dedicate future experiments to the exploration of conditions that possibly favor GlnK1-binding.

In principle, we have explored the ATP bound GlnK1 effects on GlnA1 activity in the activity assays (Fig. 2E) since ATP (3.6 mM) is present. GlnK1 however showed no effects on GlnA1 activity.

In general, the manuscript is poorly written, with grammatically incorrect sentences that at times, which stands in the way of passing on the message of the manuscript.Particular points:(1) It is mentioned that 2-OG induces the active oligomeric (dodecamer, 12-mer) state of GlnA1 without detectable intermediates. However, only 62 % of the starting inactive enzyme yields active 12-mers. Note that this is contradicted in line 212.

Thanks for pointing out this discrepancy. After removing all 2-OG as we did before MP-experiments, GlnA1 doesn’t reach full dodecamers anymore when 2-OG is re-added. This is not because the 2-OG amount is not enough to trigger full assembly, but because the protein is much more unstable in the absence of 2-OG, so we predict that some GlnA1 breaks during dialysis. See also answer reviewer #2 (1) and Fig. 1 - Fig. Suppl. 2C.

Is there any protein precipitation upon the addition of 2-OG? Is all protein being detected in the assay, meaning, is monomer/dimer + dodecamer yields close to 100% of the total enzyme in the assay?

There is no protein precipitation upon the addition of 2-OG, indeed, GlnA1 is much more stable in the presence of 2-OG. In the mass photometry experiments, all particles are measured, precipitated protein would be visible as big entities in the MP.

Please add to Figure 1 the amount of monomer/dimer during titration. Some debate why there is no full conversion should be tentatively provided.

We agree with the reviewer and included the amount of monomer/dimer in the figure, as well as some discussion on why it is not fully converted again. GlnA1 is unstable without 2-OG and it was dialysed against buffer without 2-OG before MP measurements. This sample mistreatment resulted in no full re-assembly after re-adding 2-OG although full dodecamers before dialysis (Fig. 1 - Fig. Suppl. 2C).

(2) Figure 1B reflects an exemplary result. Here, the addition of 0.1 mM 2-OG seems to promote monomer to dimer transition. Why was this not studied in further detail? It seems highly relevant to know from which species the dodecamer is assembled.

We thank the reviewer for their comment. However, we would like to point out that, although not shown in the figure, GlnA1 is always mainly present as dimers as the smallest entity. As suggested earlier, we have added the amount of monomers/dimers to Figure 1A, which shows low monomer-counts at all 2-OG concentrations (Fig.1A). Although not depicted in the graph starting at 0.01 mM OG, we also see mainly dimers at 0 mM 2-OG.

How does the y-axis compare to the number and percentage of counts assigned to the peaks? In line 713, it is written that the percentage of dodecamer considers the total number of counts, and this was plotted against the 2-OG concentration.

We thank the reviewer for addressing this unclarity. Line 713 corresponds to Figure 1A, where we indeed plotted the percentage of dodecamer against the 2-OG-concentration. Thereby, the percentage of dodecamer corresponds to the percentage calculated from the Gaussian Fit of the MP-dodecamer-peak. In Figure 1 B, however, the y-axis displays the relative amount of counts per mass, multiple similar masses then add up to the percentage of the respective peak (Gaussian Fit above similar masses).

(3) Lines 714 and 721 (and elsewhere): Why only partial data is used for statistical purposes?

We in general only show one exemplary biological replicate, since the quality of the respective GlnA1 purification sometimes varied (maximum activity ranging from 5 - 10 U/mg). Therefore, we only compared activities within the same protein purification. For the EC50 calculations of all measurements, we refer to the supplement.

(4) Lines 192-193: It is claimed that GlnK1 was previously shown to both regulate the activity of GlnA1 and form a complex with GlnA1. Please mention the ratio between GlnK1 and GlnA1 in this complex.

We now included the requested information (GlnA1:GlnK1 1:1, (Ehlers et al. 2005); His_6_-GlnA1 (0.95 μM), His_6_-GlnK1 (0.65 μM); 2:1,4, Gutt et al. 2021).

It is also known that PII proteins such as GlnK1 can bind ADP, ATP, and 2-OG. Interestingly, however, for various described PII proteins, 2-OG can only bind after the binding of ATP.So, the crucial question here is what is the binding state of GlnK1?Were these assays performed in the absence of ATP? This is key to fully understand and connect the results to the previous observations. For example, if the GlnK1 used was bound to ADP but not to ATP, then the added 2-OG might indeed only be able to affect GlnA1 (leading to its activation/oligomerization). If this were true and according to the data reported, ADP would prevent GlnK1 from interacting with any oligomeric form of GlnA1. However, if GlnK1 bound to ATP is the form that interacts with GlnA1 (potentially validating previous results?) then, 2-OG would first bind to GlnK1 (assuming a higher affinity of 2-OG to GlnK1), eventually causing its release from GlnA1 followed by binding and activation of GlnA1.These experiments need to be done as they are essential to further understand the process. Given the ability of the authors to produce the protein and run such assays, it is unclear why they were not done here. As written in line 203, in this case, "under the conditions tested" is not a good enough statement, considering what is known in the field and how many more conclusions could easily be taken from such a setup.

Thanks for the encouragement to investigate the ligand-bound states of GlnK1. We agree and plan to perform the suggested mass photometry experiments exploring the conditions under which GlnA1 and GlnK1 might interact in future work. In GlnA1 activity test assays, when evaluating the presence/effects of GlnK1 on GlnA1 activity, however, ATP was always present in high concentrations and still we did not observe a significant effect of GlnK1 on the GlnA1 activity.

(5) Figure 2D legend claims that the graphic shows the percentage of dodecameric GlnA1 as a function of the concentration of 2-OG. This is not what the figure shows; Figure 2D shows the dodecamer/dimer (although legend claims monomer was used, in line 732) ratio as a function of 2-OG (stated in line 736!). If this is true, a ratio of 1 means 50 % of dodecamers and dimers co-exist. This appears to be the case when GlnK1 was added, while in the absence of GlnK1 higher ratios are shown for higher 2-OG concentration implying that about 3 times more dodecamers were formed than dimers. However, wouldn´t a 50 % ratio be physiologically significant?

We apologize for the partially incorrect and also misleading figure legend and corrected it. Indeed, the ratio of dodecamers and dimers is shown. Furthermore, we did not use monomeric GlnA1 (the smallest entity is mainly a dimer, see Fig 1A), however, the molarity was calculated based on the monomer-mass. Concerning the significance of the difference between the maximum ratio of GlnA1 and GlnK1: The ratio does appear higher, but this is mostly because adding large quantities of GlnK1 broadens all peaks at low molecular weight. This happens because the GlnK1 signal starts overlapping with the signal from GlnA1, leading to inflated GlnA1 dimer counts. We therefore do not think that this is biologically significant, especially as the activities do not differ under these conditions.

(6) Is it possible that the uncleaved GlnA1 tag is preventing interaction with GlnK1? This should be discussed.

This is of course a very important point. We however realized that Schumacher et al. also used an N-terminal His-tag, so we assume that the N-terminal tag is not hampering the interaction.

(7) Line 228: Please detail the reported discrepancies in rmsd between the current protein and the gram-negative enzymes.

The differences in rmsd between our *M.mazei* GlnA1 structure and the structure of gram-negative enzymes is caused by (a) sequence similarity: E.g. *M.mazei* GlnA1 compared to *B.subtilis* GlnA have a sequence percent identity of 58.47; (b) ligands in the structure: The *B.Subtilis* structure contains L-Methionine-S-sulfoximine phosphate, a transition state inhibitor, while the *M. mazei* structure contains 2OG; (c) Methodology: The structural determination methods also contribute to these differences. *B. subtilis* GlnA was determined using X-ray crystallography, while the *M. mazei* GlnA1 structure was resolved using Cryo-EM, where the protein behaves differently in ice compared to a crystal.

(8) Line 747: The figure title claims "dimeric interface" although the manuscript body only refers to "hexameric interface" or "inter-hexamer interface" (line 224). Moreover, the figure 4 legend uses terms such as vertical and horizontal dimers and this too should be uniformized within the manuscript.

Thank you for your valuable feedback. We have updated both the figure title and the figure legend as well in the main text to ensure consistency in the description.

(9) Line 752: The description of the color scheme used here is somehow unclear.

Thanks for pointing this out. We changed the description to make it more comprehensive.

(10) Please label H14/15 and H14´/H15´in Fig 4C zoom.

We agree that this has not been very clear. We added helix labels.

(11) In Figure 4D legend, make sure to note that the binding sites for the substrate are based on homologies with another enzyme poised with these molecules.The same should be clear in the text: sites are not known, they are assumed to be, based on homologies (paragraph starting at line 239).

Concerning this comment we want to point out that we studied the exact same enzyme as the Schumacher group, except that we used 2-OG in our experiments, which they did not.

(12) Figure 3 appears redundant in light of Figure 4.(13) Line 235: When mentioning F24, please refer to Figure 5.

Thank you, we changed that accordingly.

(14) Please provide the distances for the bonds depicted in Figure 4B.

Thanks for pointing this out, we added distance labels to Figure 4B. For reasons of clarity only to three H-bonds.

(15) Line 241: D57 is likely serving to abstract a proton from ammonium, what is residue Glu307 potentially doing? The information seems missing in light of how the sentence is built.

Thanks for pointing this out. According to previous studies both residues are likely involved in proton abstraction - first from ammonium, and then from the formed gamma-ammonium group. Additionally, they contribute in shielding the active site from bulk solvent to prevent hydrolysis of the formed phospho-glutamate.

(16) Why do the authors assume that increased concentrations of 2-OG are a signal for N starvation only in M. mazei and not in all prokaryotic equivalent systems (line 288)?

In line 288, we did not claim that this is a unique signal for *M. mazei*. It is also the central N-starvation signal in Cyanobacteria but not directly perceived by the cyanobacterial GS through binding directly to GS.

The authors should look into the residues that bind 2-OG and check if they are conserved in other GS. The results of this sequence analysis should be discussed in line with the variable prokaryotic glutamine synthetase types of activity modulation that were exposed in the introduction and Figure 7.

Please refer to Fig. 7 - Fig. Suppl. 1, where we already aligned the mentioned glutamine synthetase sequences. Since this was also already discussed in Müller et al. 2024, we did not want to repeat their observations and refer to our supplementary figure in too much detail.

(17) Figure 5 title: Replace TS by transition state structures of homology enzymes, or alike.

Thank you for this suggestion. We did not change the title however, since it is not a homologue but the exact same glutamine synthetase from *Methanosarcina mazei*.

(18) Line 249: D170 is not shown in Figure 5A or elsewhere in Figure 5.

Thank you for pointing this out. We added D170 to figure 5A.

(19) Representative density for the residues binding 2-OG should be provided, maybe in a supplemental figure.

Thank you for the suggestion. We added the densities of 2-OG-binding residues to figure 4B

(20) Line 260: Please add a reference when describing the phosphoryl transfer.

We thank the reviewer for this important point and added that accordingly.

(21) Line 296: The binding of 2-OG indeed appears to be cooperative, such that at concentrations above its binding affinity to the protein, only dodecamers are seen (under experimental conditions). However, claiming that the oligomerization is fast is not correct when the experimental setup includes 10 minutes of incubation before measurements are done. Please correct this within the entire manuscript.A (fast) continuous kinetic assay could have confirmed this point and revealed the oligomerization steps and the intermediaries in the process (maybe monomer/dimers, then dimers/hexamers, and then hexamers/dodecamers). Such assays would have been highly valuable to this study.

We thank the reviewer for this suggestion, but disagree. It is indeed a rather fast regulation (as activity assays without pre-incubation only takes 1 min longer to reach full activity, see the newly included Fig. 1 - Fig. Suppl. 4). Considering other regulation mechanisms like e.g. transcription or translation regulation, an activation that takes only 60 s is actually quite quick.

(22) Line 305 (and elsewhere in the manuscript): the authors state that 2-OG primes the active site for a transition state. This appears incorrect. The transition state is the highest energy state in an enzymatic reaction progressing from substrate to product. Meaning, the transition state is a state that has a more or less modified form of the original substrate bound to the active site. This is not the case.In line 366 an "active open state" appears much more adequate to use.

We agree and changed accordingly throughout the manuscript.

(23) Line 330: Please delete "found". Eventually replace it with "confirmed": As the authors write, others have described this residue as a ligand to glutamine.

Thanks, we changed that accordingly, although previous descriptions were just based on homologies without the experimental validation.

(24) The discussion in at various points summarizing again the results. It should be trimmed and improved.(25) Line 381: replace "two fast" with "fast"?

We thank the reviewer for this suggestion, but disagree on this point. We especially wanted to highlight that there are **two** central nitrogen-metabolites involved in the direct regulation of GlnA1, that means TWO fast direct processes mediated by 2-OG and glutamine.